# 'Rewritable' and 'liquid-specific' recognizable wettability pattern

Manideepa Dhar[1], Debasmita Sarkar[1], Avijit Das[1], S. K. Asif Rahaman[1], Dibyendu Ghosh[1] & Uttam Manna ®[1,2,3] ✉

Bio-inspired surfaces with wettability patterns display a unique ability for liquid manipulations. Sacrificing anti-wetting property for confining liquids irrespective of their surface tension ($\gamma_{LV}$), remains a widely accepted basis for developing wettability patterns. In contrast, we introduce a 'liquid-specific' wettability pattern through selectively sacrificing the slippery property against only low $\gamma_{LV}$ (<30 mN m$^{-1}$) liquids. This design includes a chemically reactive crystalline network of phase-transitioning polymer, which displays an effortless sliding of both low and high $\gamma_{LV}$ liquids. Upon its strategic chemical modification, droplets of low $\gamma_{LV}$ liquids fail to slide, rather spill arbitrarily on the tilted interface. In contrast, droplets of high $\gamma_{LV}$ liquids continue to slide on the same modified interface. Interestingly, the phase–transition driven rearrangement of crystalline network allows to revert the slippery property against low $\gamma_{LV}$ liquids. Here, we report a 'rewritable' and 'liquid-specific' wettability pattern for high throughput screening, separating, and remoulding non-aqueous liquids.

Taking inspiration from the desert beetle, different wettability patterns have been introduced in the past for a wide range of prospective applications including moisture management[1–3], water harvesting[4–7], energy conversion[8–11], microfluidics[12–14], cell on a chip[15–17], etc. For example, nano/micro roughened superhydrophobic interfaces that repel water droplets with water contact angle >150° are commonly subjected to the localized modulation of either chemistry and (or) topography for spatially selective compromise of extreme water repellence. This creates a superhydrophilic (water contact angle <10°) domains on a superhydrophobic background— which is widely recognized as superhydrophilic–superhydrophobic patterned interfaces[18–27]. Such interfaces mostly fail to perform against non-aqueous solvents. In the recent past, superomniphobic and omniphobic slippery coatings that allow effortless rolling and sliding of both low (e.g., decane, $\gamma_{LV}$ = 23.7 mN m$^{-1}$) and high (e.g., water, $\gamma_{LV}$ = 72 mN m$^{-1}$) surface tension liquids, are mostly exposed to contact (e.g., polymerization of dopamine) and non-contact (e.g., oxygen plasma) based localized sacrifice of inert fluorinated chemistry for developing superomniphilic–superomniphobic and omniphilic–omniphobic slippery patterns[17,28–35]. Such localized modification processes led to a 'complete' compromise of the antiwetting property against both low- and high-$\gamma_{LV}$ liquids. Eventually, reported wettability patterns performed very similarly against both aqueous and non-aqueous liquids[17,28–33]. However, distinct performance (i.e., wetting/antiwetting) of the same wettability pattern against low and high surface tension liquids requires a solvent-specific localized alteration of antiwetting property but it remains a challenging task to achieve.

Another important aspect is the restoration of essential surface free energy on the spatially compromised region on the omniphobic-slippery/superomniphobic background remains an extremely challenging task to achieve—and eventually, most of the reported wettability patterns that performed against non-aqueous liquids remained irreversible in nature[17,28–34]. In a seminal report, Kota and co-workers introduced an extremely rare design of a metamorphic–superomniphobic interface for reversible perturbing

[1]Department of Chemistry, Indian Institute of Technology-Guwahati, Guwahati, Assam 781039, India. [2]Centre for Nanotechnology, Indian Institute of Technology-Guwahati, Guwahati, Assam 781039, India. [3]Jyoti and Bhupat Mehta School of Health Science & Technology, Indian Institute of Technology-Guwahati, Guwahati, Assam 781039, India. ✉e-mail: umanna@iitg.ac.in

of an essential reentrant topography to compromise embedded superoleophobicity and restoring its native property back. This structural perturbation partially affected the embedded non-adhesive superhydrophobicity and transformed it into adhesive-superhydrophobicity[34]. While the fabrication of liquid-specific wettability patterns on an omniphobic background is difficult to achieve, we aim to develop a chemically reactive omniphobic slippery interface to attain (i) 'liquid-specific' and (ii) 'rewritable' wettability pattern. The ability to rewrite different wettability patterns on the same interface multiple times is a sustainable approach to a post-synthetic reconfiguration of the architectures of wettability patterns[21,34,36].

Here, we report a strategy to derive a non-fluorinated, chemically reactive and omniphobic solid slippery interface by infusing a crystalline network of a comb-like thermoplastic random copolymer, i.e., polyoctadecyl acrylate-co-maleic anhydride (PODAMA, Fig. 1a–c) in a hydrophilic porous matrix. The crystalline packing of octadecyl moiety of PODAMA provided a solid interface with depleted roughness (-22 ± 3.5 nm) and low surface free energy (-22.2 ± 0.6 mN m$^{-1}$) that can repel liquids with a wide range of surface tension from 22.7 to 72 mN m$^{-1}$ (Fig. 1a–c). Further, available amine-reactive maleic anhydride moiety in this crystalline network provides a versatile platform for desired chemical alteration by facilitating the post-covalent modifications through a ring-opening nucleophilic addition reaction at ambient conditions. The modification of the chemically reactive crystalline surface with a hydrophilic amine, i.e., glucamine moderately elevated the surface free energy to -32 mN m$^{-1}$ and selectively compromised the slippery property against only low (< 30 mN m$^{-1}$) surface tension liquids—and resulted their arbitrary spillage on the tilted interface. However, droplets of high surface tension liquids (> 35 mN m$^{-1}$; Fig. 1e) still remained efficient to slide on the same modified interface. Furthermore, subjecting this glucamine-modified surface at elevated temperature (> 55 °C) led to the melting of infused PODAMA, which disrupted the crystalline packing, facilitating migration and conformational changes in the infused PODAMA. The reorganized and melted PODAMA solidified and recrystallized on cooling to room temperature, exposing the residual maleic anhydride groups at the interface, and so resulting in a low surface free energy state.

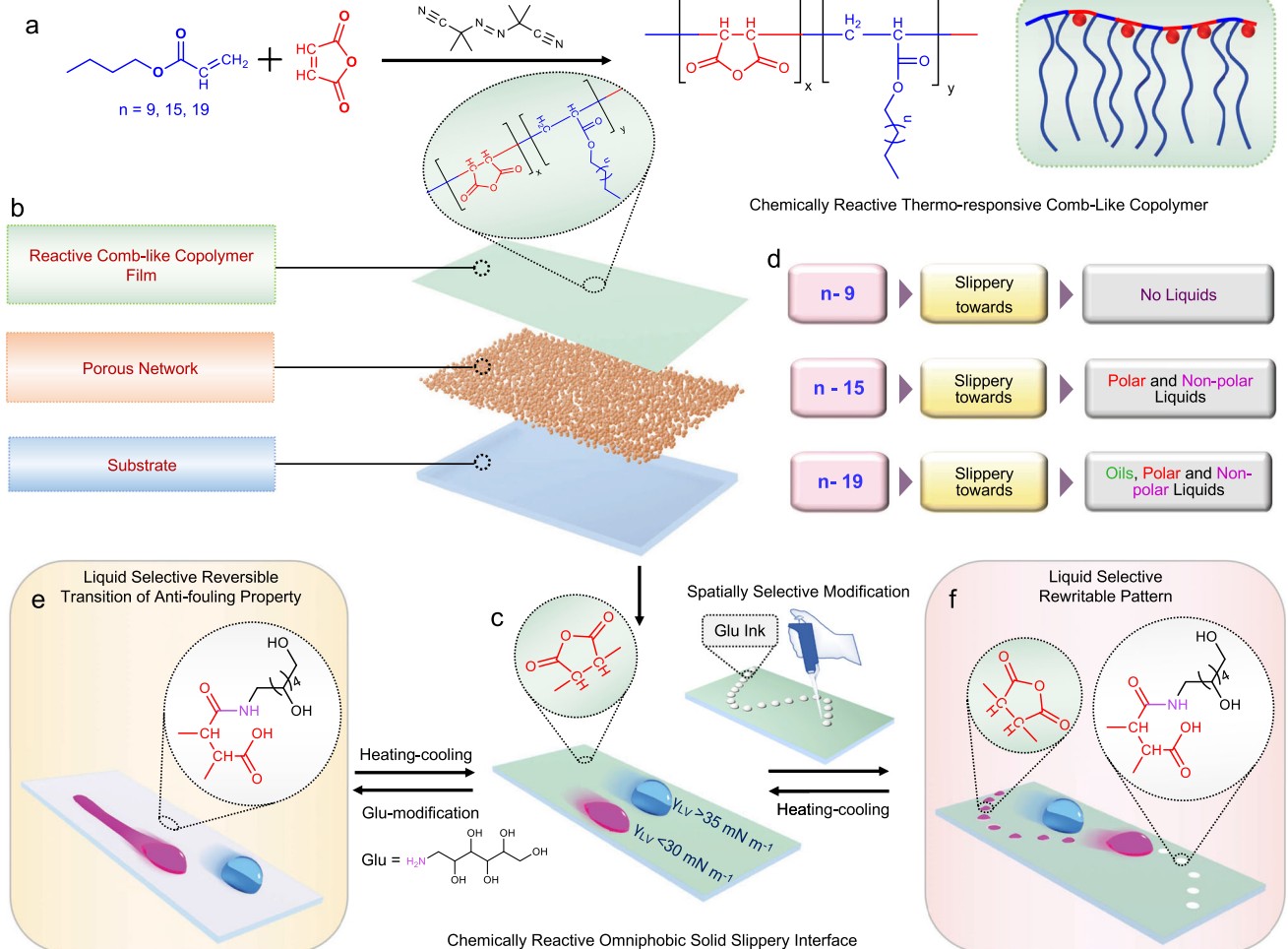

**Fig. 1 | Illustration for the fabrication of 'rewritable' and 'liquid-specific' recognizable wettability pattern.** **a** Schematic representation of the synthesis of chemically reactive thermo-responsive comb-like random copolymers, where the content of maleic anhydride and the hydrocarbon chain length of alkyl acrylate (*n* = 9, 15, 19) were systematically varied. **b, c** Schematic depicting the preparation of chemically reactive omniphobic solid slippery interface (**c**) through the infusion of reactive comb-like polymer into a porous matrix. **d** Illustrating the impact of hydrocarbon chain length on the slippery property towards a range of liquids—including aqueous and non-aqueous liquids. **e** Demonstrating the selective and reversible alteration of antiwetting property towards only low surface tension liquids through strategic post-covalent modification of prepared interface with hydrophilic small molecule—i.e., glucamine, where the melting of used comb-like copolymer allows to restore the embedded omniphobic slippery property. **f** Construction of liquid selective recognizable and rewritable wettability pattern through strategic and spatially selective modification of the coated interface with glucamine—followed by thermal treatment above the melting point of used polymer.

Eventually, the surface regained its reactive omniphobic slippery property with the ability to undergo repeated chemical modification for selective sacrifice of antiwetting property against low surface tension liquids, showcasing the reversibility of the process (Fig. 1e, c). This finding was capitalized to develop a rewritable 'liquid-specific' wettability pattern by adopting the same chemical modification—but in a spatially selective manner, with the ability to successfully reconfigure the same patterned interface multiple times. (Fig. 1f, c).

## Results

### Designing healable and reactive omniphobic slippery coating

Bio-inspired fabrication of superomniphobic interfaces commonly required a very stringent optimization of topography—i.e., reentrant structures topped with mostly fluorinated chemicals[34,37,38]. In contrast, as an alternative approach, different strategies have been introduced to develop smooth and solid slippery interfaces for effortless sliding of beaded droplets of both low and high surface tension liquids[39–48]. Among available reported designs of slippery coating, the approach of infusing non-fluorinated and chemically inert phase transitioning agents—i.e., wax (paraffin wax) or comb-like polymer (polyoctadecyl acrylate) in a porous matrix has been followed to prepare a solid slippery interface with a unique ability to heal severe physical damages[46–48]. However, such chemically inert interfaces remained inappropriate for further post-chemical modification and hence the customization of liquid antiwetting property of such reported slippery interfaces is an extremely challenging task to achieve[39–48]. Inspired by such reports of healable solid slippery coatings[46–48], we attempted to design a rare omniphobic solid slippery coating that would remain healable and chemically reactive for post-covalent modification at ambient conditions. In this context, a readily amine reactive crystalline comb-like random copolymer, i.e., polyoctadecyl acrylate-co-maleic anhydride (PODAMA) was synthesized by solution radical polymerization of selected monomers, i.e., octadecyl acrylate (ODAc) and maleic anhydride (MA) at different molar ratios—to gradually vary the content of amine-reactive moiety i.e., maleic anhydride in the prepared copolymer. Four different random copolymers were synthesized and denoted as PODAMA$_1$, PODAMA$_2$, PODAMA$_3$, and PODAMA$_4$; where the content of MA was gradually increased from PODAMA$_1$ to PODAMA$_4$ (mole fraction of MA in PODAMA$_1$, PODAMA$_2$, PODAMA$_3$ and PODAMA$_4$ is estimated to be 0.13, 0.27, 0.48 and 0.59 respectively, Supplementary Fig. 1). These synthesized copolymers (PODAMA$_{1/2/3/4}$) were characterized using attenuated total reflectance-Fourier transform infrared spectroscopy (ATR-FTIR) and nuclear magnetic resonance (NMR) spectra analysis, as shown in Fig. 2a and Supplementary Fig. 1. In the ATR-FTIR spectra, the disappearance of the IR peak at 1410 cm$^{-1}$ corresponding to vinylic C–H stretching and the appearance of two new peaks at 1846 cm$^{-1}$ and 1782 cm$^{-1}$ corresponds to carbonyl group (C=O) stretching vibrations of anhydride group indicates the successful radical copolymerization of selected monomers, i.e., ODAc with MA. As expected, the peak intensity at 1846 cm$^{-1}$ and 1782 cm$^{-1}$ correspond to carbonyl group (C=O) stretching vibrations of the anhydride group enhanced with increasing the content of MA moiety in these different copolymers (Fig. 2a), where the IR signals were normalized with respect to an invariable characteristics IR signal at 1730 cm$^{-1}$ corresponds to an ester carbonyl group (C=O) stretching. $^1$H NMR spectroscopy further supports the successful synthesis of these copolymers (PODAMA$_{1/2/3/4}$). The peak integration of $^1$H NMR signal at 3.7 ppm corresponding to the -CH group of maleic anhydride moiety increases gradually with increasing the amount of MA moiety in different copolymers (PODAMA$_{1/2/3/4}$) (Supplementary Fig. 1). It is worth mentioning that loaded MA moiety in PODAMA readily and mutually reacts with primary amine-containing small molecules. The successful chemical modification of PODAMA with selected small molecules i.e., glucamine was characterized with ATR-FTIR analysis (Supplementary Fig. 2). The successful occurrence of the ring-opening reaction was confirmed by the appearance of two new peaks at 1641 cm$^{-1}$ and 1564 cm$^{-1}$, corresponding to amide-I and amide-II, respectively. Simultaneously, a reduction in characteristics IR peaks at 1846 cm$^{-1}$ and 1782 cm$^{-1}$, associated with the carbonyl group of the maleic anhydride ring, revalidated the existence of mutual ring-opening reaction.

Thereafter, such chemically reactive polymers (PODAMA$_{1/2/3/4}$) were infused in a porous matrix to derive a reactive polymer-infused coating (RPIC). In this context, a hydrophilic porous matrix (Supplementary Fig. 3) was developed by spray depositing the reaction mixture of branched polyethylenimine (BPEI) and dipentaerythritol pentaacrylate (5-Acl). The prepared porous matrix enabled the infusion of synthesized chemically reactive polymers (PODAMA$_{1/2/3/4}$) to get a smooth, reactive, and crystalline interface. All the reactive polymer-infused coatings (RPIC$_{1/2/3/4}$) remained capable of forming a crystalline network as confirmed by the appearance of a characteristic X-ray diffraction (XRD) peak at 2θ of 22°. These peaks correspond to the hexagonal close packing of octadecyl side chains of prepared copolymers and resemble the crystalline α phase (hexagonal packing) of ODAc monomer (Fig. 2b). However, these coatings (RPIC$_{1/2/3/4}$) derived from different copolymers (PODAMA$_{1/2/3/4}$: loaded with different amounts of MA moiety) displayed a significant contrast in liquid wettability and antiwetting property towards both high (e.g., water, $\gamma_{LV}$ = 72 mN m$^{-1}$) and low (e.g., ethanol, $\gamma_{LV}$ = 22.3 mN m$^{-1}$) surface tension liquids (Fig. 2c, d and Supplementary Fig. 4). For example, droplets of water and ethanol beaded on RPIC$_2$ with static contact angles (SCA) of ~112 ± 0.8° and ~33 ± 0.6°, respectively and effortlessly slide (sliding angles of 20° and 12° for water and ethanol, respectively) without leaving behind any traces (Fig. 2c). However, prepared RPIC$_4$ failed to display antiwetting property against low surface tension liquid i.e., ethanol, rather it beaded with low SCA (~19 ± 0.8°) and arbitrarily spread on tilting the interface. Nevertheless, beaded water droplets having high surface tension continued to slide on such interface at a little elevated tilting angle (30° ± 2°) with SCA of 88 ± 0.6°, highlighting a hydrophilic solid slippery property of RPIC$_4$ (Fig. 2d).

To understand such distinct antiwetting performance exhibited by different RPIC, their surface morphologies were examined, where the mole fraction of MA moiety in selected polymers (PODAMA$_{1/2/3/4}$) was gradually increased from 0.13 to 0.59. Atomic force microscopic (AFM) images validated the difference in the topography of prepared coatings (Supplementary Fig. 5). While RPIC$_2$ was embedded with dominated fibril domains (Fig. 2e), RPIC$_4$ displayed an ultra-smooth coating (Fig. 2f). A very apparent depletion in the root mean square roughness ($R_q$) was also noticed from ~ 48 ± 3.7 nm to ~ 0.52 ± 0.1 nm with increasing the MA content in the selected polymers (PODAMA$_{1/2/3/4}$) (Fig. 2g). While the decrease in surface roughness is expected to improve the sliding behaviour of beaded droplets of liquids, we noticed a distinct trend. The polymeric coating with mole fraction of MA ≤ 0.48 displayed antiwetting properties against both high and low surface tension liquids i.e., water ($\gamma_{LV}$ = 72 mN m$^{-1}$) and ethanol ($\gamma_{LV}$ = 22.3 mN m$^{-1}$), whereas PODAMA$_4$ based coating (RPIC$_4$) having mole fraction of MA of 0.59 failed to slide ethanol droplets, as shown in Fig. 2d. Thus, the antiwetting property of these coatings against low surface tension liquids is not solely determined by surface topography alone. We examined another important parameter i.e., the surface free energy (SFE) of the same coatings to understand such differences in liquid antiwetting performance. With increasing MA content in the infused polymer, the SFE gradually elevated (Fig. 2g). We noticed that the coating (RPIC$_4$) loaded with MA 0.59 mole fraction associated with relatively high SFE of ~ 32 ± 0.2 mN m$^{-1}$ and failed to slide droplets of low surface tension liquids (e.g., ethanol) despite having an ultralow $R_q$ of 0.52 ± 0.1 nm. Whereas the other three compositions of PODAMA that provided crystalline interface (RPIC$_{1/2/3}$) with lower SFE < 25 mN m$^{-1}$ remained effective in sliding both high and low

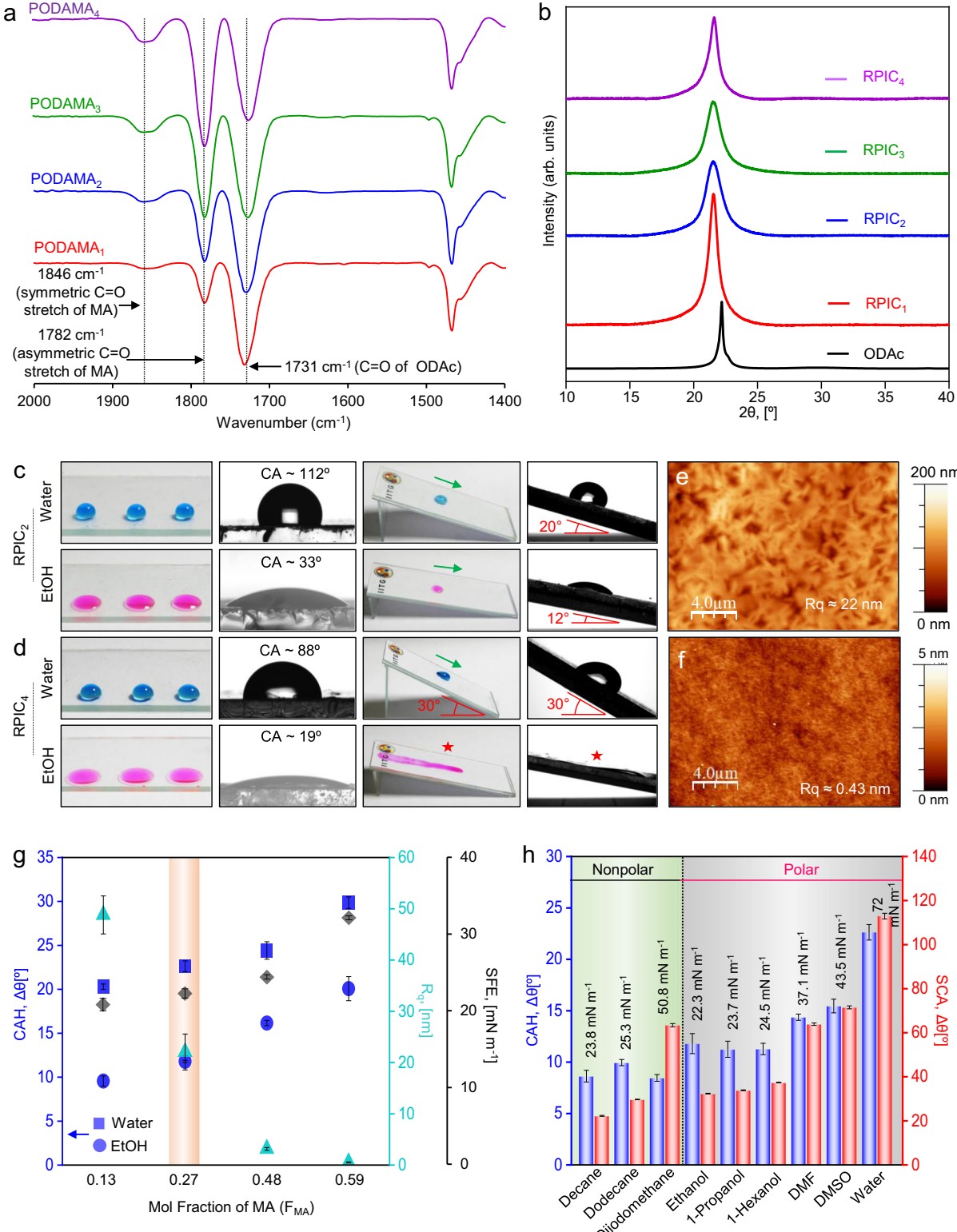

**Fig. 2 | Preparation of healable and chemically reactive omniphobic solid slippery coating. a** ATR-FTIR spectra of different random copolymers (PODAMA$_{1/2/3/4}$) having differences in the content of reactive maleic anhydride moiety. **b** X-ray diffraction (XRD) pattern of an octadecyl acrylate monomer and reactive copolymers (PODAMA$_{1/2/3/4}$) infused coatings (RPIC$_{1/2/3/4}$). **c, d** Digital and contact angle images depicting the beading and sliding or spillage of water and ethanol droplets on RPIC$_2$ (**c**) and RPIC$_4$. **e, f** Atomic force microscopic (AFM) images of RPIC$_2$ (**e**) and RPIC$_4$ (**f**). **g** Plot demonstrating the change in contact angle hysteresis (CAH; blue) of beaded water and ethanol droplet, root mean square

roughness (R$_q$; turquoise), surface free energy (SFE; black) of different copolymer-infused coatings RPIC$_{1/2/3/4}$ with the variation of maleic anhydride content. The orange highlighted region indicates the optimized mol fraction of MA used for further studies. **h** Plot of CAH and static contact angle (SCA) of different polar and non-polar liquids on RPIC$_2$. Green and grey highlighted region indicates non-polar and polar liquids, respectively. The error bar indicates the standard deviation with number of measurements, $n = 3$ for each data point. Source data are provided as a Source Data file.

surface tension liquids (Fig. 2g and Supplementary Fig. 4)—though $R_q$ of these coatings was found to be relatively higher (48 nm to 3 nm) compared to RPIC$_4$.

In the past, it has been realized that the dissipative force ($F_d$) acting on the sliding droplet depends on the contact angle hysteresis of the beaded droplet[49] (Eq. (1));

$$F_d = 2a\gamma\Delta\cos\theta \qquad (1)$$

Where $\Delta\cos\theta$ is equal to $\cos\theta_{rec} - \cos\theta_{adv}$, $\gamma$ and $a$ are the surface tension and base radius of the beaded liquid droplet, respectively.

A depleted contact angle hysteresis (CAH) of a given liquid on a slippery surface indicates the existence of a low dissipative force[49]. The CAH of water (~20° to ~30°) and ethanol (~9° to 20°) droplets increases gradually with rising more MA moiety in the infused copolymers (PODAMA$_{1/2/3/4}$) of prepared coatings (RPIC$_1$ to RPIC$_4$; Fig. 2g). The elevated SFE of the coatings because of the increasing in the content MA moiety is likely to induce more polar-polar interaction, and so higher CAH for RPIC$_4$. Thus, the current study experimentally validates that the SFE plays a superior role over roughness towards slippery property against low surface tension liquids, and CAH of the beaded liquid droplets on the prepared coating.

A separate set of experiments was performed to examine the influence of hydrocarbon chain length in the infused copolymer on the slippery property of beaded liquids. The selected random copolymer (polylauryl acrylate-co-maleic anhydride, PLAMA) having a short hydrocarbon chain (i.e., $n = 9$ (lauryl acrylate, LAc)) failed to repel both low and high surface tension liquids (Fig. 1d and Supplementary Fig. 6). Whereas, comb-like copolymers with longer hydrocarbon chain length (i.e., $n = 15$ (ODAc)) (already discussed in detail in the previous section) exhibited liquid repellence towards both low and high surface tension liquids including different polar and non-polar liquids with surface tension ranging from ~22 mN m$^{-1}$ to ~72 mN m$^{-1}$ (Fig. 2h and Supplementary Figs. 7, 8) though failed to slide beaded oil droplets. However, the interface derived from the random copolymer (polydocosyl acrylate-co-maleic anhydride, PDAMA) having docosyl moiety ($n = 19$, (docosyl acrylate, DAc)) displayed the ability to slide even commercially available refined oils and crude oil along with other low and high surface tension organic liquids (Fig. 1d and Supplementary Fig. 9, 10). Thus, overall chemistry played a dominating role over topography in achieving slippery property across a wide range of liquids. It is worth mentioning that other bio-inspired oil/oily liquid repellence coating i.e., super-omniphobic interfaces are known to have strict requirements for topography[20,37,38].

Taking account of the optical transparency of all these developed coatings (Supplementary Fig. 11) and slippery behaviour towards both low and high surface tension liquids, solid slippery coating (RPIC$_2$) with a thickness of ~20 ± 1.6 µm prepared by infusing PODAMA$_2$ (having 0.27-mole fraction of MA, M$_n$ ≈ 8200 Da) into a porous matrix (Supplementary Fig. 12) was selected for further studies. Interestingly, the prepared RPIC$_2$ displayed an inherent ability to heal severe physical damages including deep scratches at elevated temperatures due to the phase–transition behaviour of infused polymer at ~52 °C (Supplementary Fig. 13). Above this phase–transition temperature, the prepared crystalline network of the polymer melted to cure the severe physical damage. On further cooling down to room temperature, the molten polymer recrystallized and regained the native sliding property against both low and high surface tension liquids (Supplementary Fig. 14). Thus, we developed an omniphobic solid slippery coating loaded with chemically reactive moiety i.e., MA, which remained capable of healing physical damages and wettability.

## Liquid-specific, reversible change in omniphobic slippery property

Here, we capitalize the predominated influence of SFE on the association of omniphobic slippery property against low surface tension liquids. As the prepared omniphobic slippery interface is loaded with residual reactivity, we aimed to perturb the SFE to demonstrate selective compromise of the antiwetting performance against low surface tension liquids, while keeping the antiwetting property towards high surface tension liquids intact, as shown in Fig. 3a. In this context, RPIC$_2$, which effortlessly slide beaded droplets of both ethanol and water (Fig. 3b), was submerged in a solution of a primary amine group containing hydrophilic small molecule—glucamine (denoted as Glu) to initiate the reaction between surface available MA moiety of RPIC$_2$ with the primary amine group of Glu through ring-opening reaction at ambient condition. This chemical treatment of RPIC$_2$ significantly altered the antiwetting property of the coating, where the droplet of ethanol beaded with a depleted SCA (~19 ± 0.7°) and failed to slide on the modified interface. Instead, arbitrary spilling of ethanol on the modified coating was noted, as shown in Fig. 3c and Supplementary Fig. 15. However, the droplet of water continued to slide on such interface with low (< 90°) SCA (Fig. 3c and Supplementary Fig. 15). In fact, after chemical modification of RPIC$_2$ with glucamine, beaded droplets (10 µl) of other liquids of low (< 30 mN m$^{-1}$) surface tension, including decane, dodecane, ethanol, 1-propanol and 1-hexanol failed to slide, instead the beaded droplets of these liquids arbitrarily spilled on the Glu-modified RPIC$_2$ (Supplementary Fig.16b). Whereas, on contrary, droplets of other liquids having surface tension >35 mN m$^{-1}$ continued to slide on the same modified interface (Supplementary Fig. 17b). However, prior to its post-covalent modification, droplets of both low (<30 mN m$^{-1}$) and high surface tension liquids effortlessly slide on the tilted interface of RPIC$_2$ as shown in Supplementary Figs. 16a, 17a. Thus, such chemical modification-based alteration of wetting behaviour from sliding to spilling was only observed for beaded droplets of liquids having surface tension < 30 mN m$^{-1}$ (Supplementary Figs. 16–18). Moreover, such selective spilling of liquid droplets (surface tension < 30 mN m$^{-1}$) was observed over a wide range of tilting angles of the modified interface (Supplementary Fig. 19). Also, beaded droplets of liquids having surface tension <30 mN m$^{-1}$ were found to be spread down the slope, irrespective of the volume of selected liquids (Supplementary Fig. 20). Thus, the prepared interface would be appropriate for real-world applications, where the optimization of the tilting angle of surface is not required to monitor the selective spillage of beaded liquids of surface tension <30 mN m$^{-1}$. Thereafter, the surface modification of this crystalline network with glucamine was characterized with an X-ray photoelectron spectroscopic (XPS) spectral study. The deconvoluted XPS spectrum {C($1s$)} of RPIC$_2$ reveals the presence of five types of carbon functionalities, including, hydrocarbon (CHx: 284.7 eV), carbon singly bonded to an anhydride group (C–C(O)=O: 285.3 eV), carbon singly bonded to oxygen (C–O: 285.9 eV), carbon doubly bonded to oxygen (O–C–O/C=O: 287 eV), and anhydride (O=C-O-C=O: 289.3 eV) (Fig. 3d)[50]. A change in the C($1s$) envelope was observed after the treatment of RPIC$_2$ with glucamine—and an appearance of characteristic signature for the amide group (RNH–C=O: 288.7 eV) confirmed the successful surface modification of the coating with glucamine through ring-opening reaction (Fig. 3d)[50]. Additionally, ATR-FTIR spectral analysis was also performed on RPIC$_2$ before and after modification with glucamine to revalidate the post-covalent modification on the surface. The depletion of IR peak intensities at 1846 cm$^{-1}$ and 1782 cm$^{-1}$, corresponding to carbonyl group (C=O) stretching vibrations of the anhydride group with respect to the invariable internal reference, i.e., IR signal at 1730 cm$^{-1}$ for the ester carbonyl group (C=O) stretching independently validated the successful modification of RPIC$_2$ with glucamine. In addition, two new IR peaks related to amide-I and amide-II appeared at 1641 cm$^{-1}$ and 1564 cm$^{-1}$ which again supported the reaction of

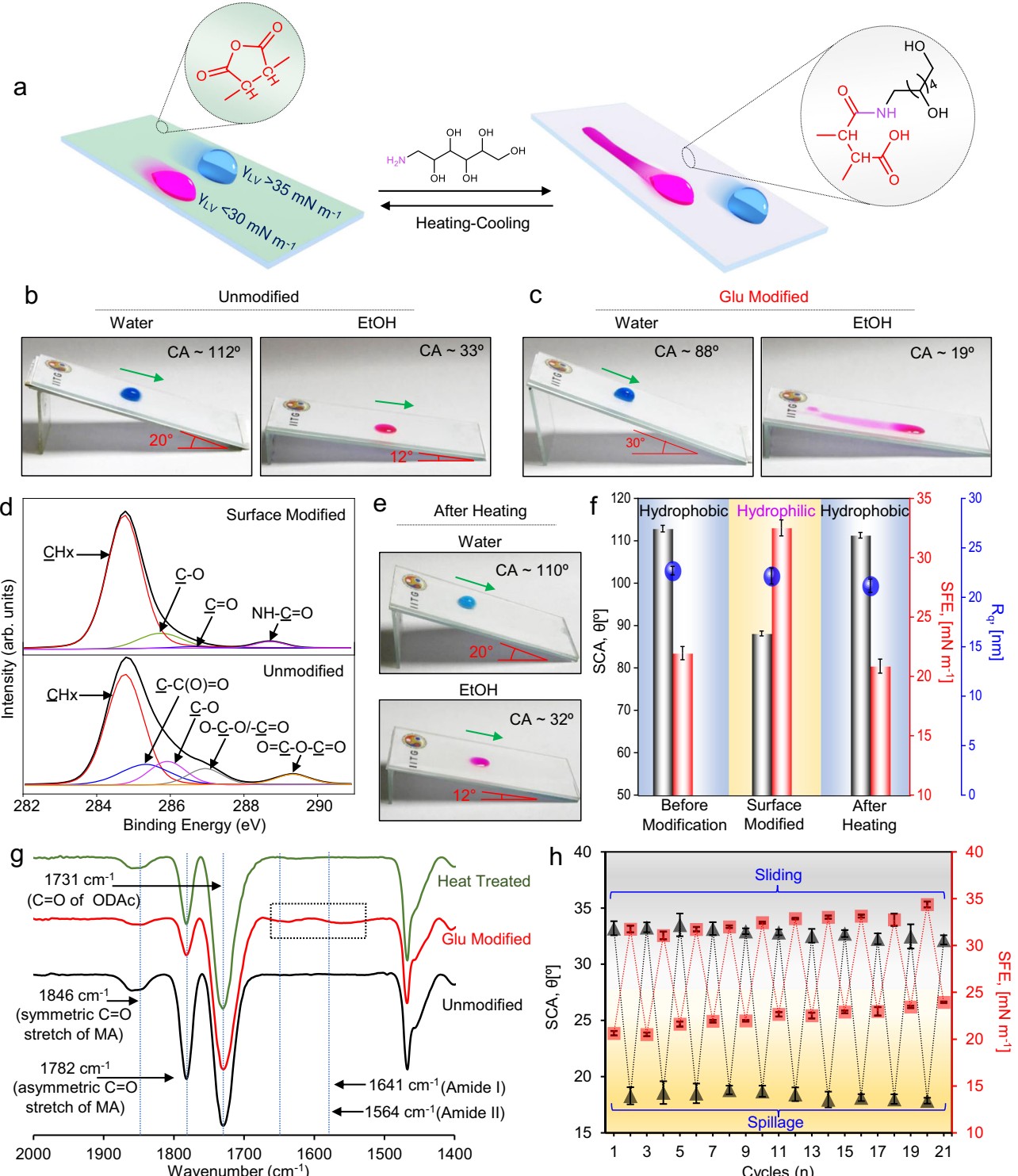

**Fig. 3 | Reversible transition of liquid selective antiwetting property of prepared coating. a** Schematic demonstrating the reversible sacrifice of antiwetting property against low surface tension liquids achieved through strategic ring-opening reaction of prepared coating with glucamine, whereas association of thermal treatment reverts back the compromised antiwetting property against low surface tension liquids. **b, c** Photographs depicting the sliding or spreading of beaded water and ethanol droplets on RPIC$_2$ before (**b**) and after (**c**) glucamine modification. **d** High-resolution XPS spectra of C($1s$) measured on unmodified and glucamine-modified RPIC$_2$ surface. **e** Photographs demonstrating the regaining of native liquid repellence of water and ethanol droplets by heating and subsequent cooling process of glucamine-modified RPIC$_2$ surface. **f** Plot accounting for the

changes in static contact angle (SCA; black) of water, surface free energy (SFE; red) and root mean square roughness (R$_q$; blue) of unmodified and glucamine-modified RPIC$_2$ surface−and after heat treatment (>55 °C). Blue and yellow highlighted region indicates hydrophobicity and hydrophilicity, respectively. **g** ATR-FTIR spectra of unmodified (black), glucamine-modified RPIC$_2$ surface before (red) and after (green) heat treatment. **h** Plot accounting for the alteration in static contact angle of beaded ethanol and surface free energy (SFE) of RPIC$_2$ during multiple cycles (20 cycles) glucamine modification and subsequent heat treatment. Grey and yellow highlighted region indicates sliding and spillage, respectively. The error bar indicates the standard deviation with number of measurements, $n$ = 3 for each data point. Source data are provided as a Source Data file.

available MA moiety at the crystalline surface of $RPIC_2$ with glucamine (Fig. 3g). Such post-covalent modification of $RPIC_2$ moderately enhanced the surface free energy as shown in Fig. 3f and contributed to selective compromise of antiwetting property against low surface tension liquids. No noticeable change in the roughness of $RPIC_2$ was encountered before and after post-covalent modification (Fig. 3f). Similarly, another chemically reactive comb-like polymer having a higher analogue of alkyl chain, i.e., polydocosyl acrylate maleic anhydride (PDAMA) was infused in a porous matrix to derive solid slippery coating (denoted as $RPIC_{PDAMA}$) with an ability to slide even commercially available refined oils (motor oil, vegetable oil, diesel, petrol, kerosene) and complex crude oil (Supplementary Fig. 10) in addition to polar and non-polar liquids. Such solid slippery coating was subjected to post-modification with glucamine to perturb the surface chemistry. As a consequence, droplets of oils and organic solvents with low (< 30 mN m$^{-1}$) surface tension readily spilled on the tilted interface, as shown in Supplementary Fig. 21. The post-covalent modification of $RPIC_{PDAMA}$ elevated the surface free energy of the coating from ~20 mN m$^{-1}$ to ~35 mN m$^{-1}$ (Supplementary Fig. 22). However, such chemical modification merely perturbed the surface roughness of the coating, as shown in Supplementary Fig. 22. Hence, this study suggests that the alteration of surface chemistry is sufficient to selectively compromise the sliding behaviour of low surface tension (< 30 mN m$^{-1}$) organic solvents and refined/crude oils, where alteration of topography is not required. Thus, the strategic modulation of chemistry provides a basis to efficiently alter the liquid sliding/spilling behaviour on a solid surface, and such principle of chemical modulation would allow to derive other functional materials in the future. More interestingly, the compromised antiwetting property of the coating against low surface tension liquids can be successfully restored by simply heating the modified coating above the phase–transition temperature of infused polymer, i.e., >55 °C (Supplementary Fig. 13) and subsequently cooling down to room temperature, as shown in Fig. 3e. Upon cooling down the same interface to room temperature, droplets of both low (ethanol) and high (water) surface tension liquids beaded on the surface with relatively higher SCA (Supplementary Fig. 23) and readily started sliding on tilting, as shown in Fig. 3e. This heating and cooling process depleted the SFE of the modified interface from ~32 ± 0.7 nm to ~22 ± 0.6 nm (Fig. 3f). Such depletion of SFE certainly contributed to retrieve the compromised antiwetting property. No apparent change in the roughness of the coating was noticed. Such reversible alteration of the effortless sliding and arbitrary spillage behaviour of the prepared coating against low surface tension liquid can be successfully repeated for multiple times (i.e., 20 times, Fig. 3h). To explain this phenomenon, we hypothesized; that on melting the infused polymer above its phase–transition temperature (Supplementary Fig. 13), the crystalline arrangement of the polymer is compromised and on cooling the recrystallization of the polymer allowed favourable change in the SFE. At the molten state of the polymer, migration and conformational change of the polymer are expected to happen[51,52] which is likely to bury the Glu-modified hydrophilic domains and facilitate the exposure of fresh reactive MA moiety (Fig. 4). Thus, on cooling the same coating, another recrystallized network was obtained with low surface free energy (Fig. 3f). The redistribution of functional group at the interface was realized through ATR-FTIR analysis. The normalized (with respect to the peak at 1730 cm$^{-1}$) IR peak intensities at 1846 cm$^{-1}$ and 1782 cm$^{-1}$ increased in comparison to Glu-modified coating after the heat treatment (Fig. 3g). Beyond 20 cycles, the $RPIC_2$ failed to restore the native sliding property against the low surface tension liquids due to the limited availability of fresh and adequate MA moiety in the prepared coating of thickness ~20 ± 1.6 µm, where the content of infused $PODAMA_2$ is 50 mg cm$^{-2}$. The restoration ability of native sliding property was further attempted to improve just by loading more reactive comb-polymer in the porous matrix–during its fabrication process. In this

relevance, we have loaded more $PODAMA_2$ in the porous layer to prepare thicker $RPIC_2$ for successfully demonstrating the sliding/spilling cycles beyond 20 cycles.

The higher contents (100 mg cm$^{-2}$ and 200 mg cm$^{-2}$) of $PODAMA_2$, resulted in $RPIC_2$ with a thickness of ~51 ± 1.8 µm and ~71 ± 1.4 µm and displayed reversible alteration of sliding and arbitrary spilling of low surface tension liquid on the prepared coating for 40 and 70 times, respectively as shown in Supplementary Figs. 24, 25. Actually, the higher loading of $PODAMA_2$ ensures more availability of fresh MA moiety even after repetitive surface modification and followed by its erasure beyond 20 times. Although there is a limit in terms of the number of cycles for surface modification and followed by its erasure, it can be easily adjusted by varying the amount of infused polymer, thereby altering the thickness of the prepared coatings. This demonstrates the versatility of the fabrication process–which contributes to improving the performance of the prepared slippery coating. The reversible disruption of antiwetting property selectively against low surface tension liquid through post-covalent modification with small molecules and followed by facile restoration of native liquid wettability through recrystallization of polymer network is unprecedently demonstrated here.

## Liquid-selective, rewritable wettability pattern
Unlike above-mentioned procedure where the entire top surface was post-modified with glucamine, here a liquid selective recognizable and rewritable wettability pattern is constructed through spatially selective post-covalent modification of the prepared chemically reactive omniphobic solid slippery coating with Glu, as shown in Fig. 5a, d. To prepare such pattern wettability, tiny droplets (2 µl) of Glu in dimethyl sulfoxide (DMSO) were placed on the $RPIC_2$ with a specific arrangement in the shape of 'A' for 12 h. Over time, available MA moiety on the selected top portion of the crystalline network reacted with the Glu molecule through a mutual ring-opening reaction as discussed in the earlier section. It compromised the antiwetting property against low surface tension liquids (e.g., ethanol) spatially selectively at the modified region maintaining an unperturbed omniphobic slippery background. Thus, the "A" shaped pattern on the same interface was readily recognized with the naked eye upon ethanol exposure due to spatially selective fouling by exposed ethanol on the patterned region (Fig. 5c and Supplementary Movie 1). However, the same pattern remained completely hidden towards water exposure because of the existence of unperturbed antiwetting property against a high surface tension liquid, as shown in Fig. 5b. Consequently, the same prepared pattern interface displayed a completely distinct appearance against low and high surface tension liquids. Thereafter, it was repetitively reconfigured by adopting different architectures of spatially selective chemical modifications followed by the melting of $PODAMA_2$ at its phase–transition temperature. The principle of melting of such interface > 55 °C and its recrystallization at room temperature allowed to regain compromised antiwetting property against low surface tension liquids and eventually, the wettability pattern was erased providing a basis to rewrite or reconfigure another wettability pattern on the same interface, as illustrated in Fig. 5d. As a proof-of-concept demonstration, the interface with wettability pattern in "A" shape was exposed to 70 °C for 15 min before cooling down to room temperature. After the heating/cooling cycle, the beaded droplet of ethanol (coloured with Rhodamine B) effortlessly slide off without leaving any trace of it, as shown in Fig. 5e, suggesting a complete erase of the pattern wettability. Thereafter, spatially selective chemical modification with glucamine in the shape of 'T' was performed on the same interface, resulting in a "T" shaped wettability pattern, as shown in Fig. 5e. This process can be repeated multiple times to erase and reconfigure the wettability patterns of our choice. Thus, the current approach of designing a chemically reactive omniphobic solid slippery interface

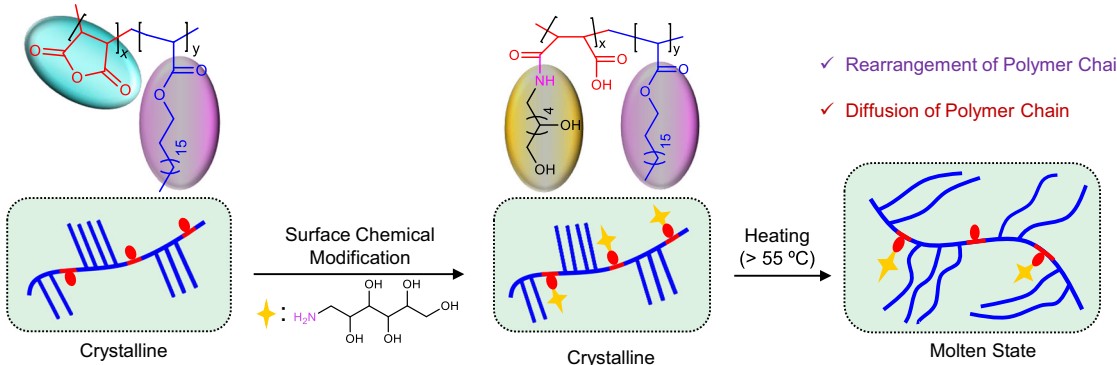

**Fig. 4 | Rearrangement of infused polymers.** Schematic demonstrating the possible rearrangement and diffusion of polymer chain at the molten state of infused PODAMA being responsible for restoring the native liquid slippery property after the heat treatment of the glucamine-modified surface.

provided a facile basis to develop a rewritable and liquid selective recognizable wettability pattern.

## Pattern wettability to separate and manipulate non-aqueous liquids

In the last set of experiments, we demonstrated the potential uses of such rewritable and liquid selective recognizable wettability patterns in high throughput sorting and separation of liquids from their mixture based on their surface tension, designing oil catcher, liquid-liquid extraction followed by targeted chemical delivery, liquid moulding for interfacial chemical reactions, etc. In this context, a wettability pattern was designed, where a continuous surface channel was made on the omniphobic slippery interface of RPIC$_2$ through spatially selective glucamine modification. This continuous surface channel remained recognizable by low surface tension liquids and allowed guided transport of such liquids from one end to the other end of the patterned interface, as shown in Fig. 6a. However, such a pattern is expected to remain unrecognizable by high surface tension liquid, as shown in Fig. 6a. As a proof-of-concept demonstration, such patterned interface was kept at a tilting angle of 12° before introducing a droplet of water at the beginning of the chemically modified channel at the top and left corner of the pattern, and the water droplet effortlessly slide down straight under gravity without recognizing the chemically modulated pattern (Fig. 6b and Supplementary Movie 2). In contrast, a beaded droplet of ethanol recognized the pattern and followed the glucamine-modified track, moving to the other end of the patterned interface (Fig. 6b and Supplementary Movie 2). Similarly, other polar and non-polar liquids with low $\gamma_{LV}$ (e.g., 1-propanol, decane, 1-hexanol and dodecane) preferred to follow the pattern region (Fig. 6c, Supplementary Fig. 26), while the high surface tension liquids [>35 mN m$^{-1}$; dimethyl formamide (DMF), DMSO and diiodomethane (DIM)] readily slide straight down on the surface as shown in Fig. 6c and Supplementary Fig. 26. The patterned interface similarly performed even at a lower tilting angle as the chemically modified interface displayed the selective spillage of only low (< 30 mN m$^{-1}$) surface tension liquids, irrespective of the tilting angle (Supplementary Fig. 19). In this relevance, the patterned interface was tilted at an angle of 5° before introducing the beaded droplets of ethanol and water separately. The low surface tension liquid, i.e., ethanol, easily recognizes the chemically modified region and selectively spilled along the chemically modified region. However, the beaded water droplet (high surface tension liquid) behaved entirely differently on the same patterned interface than the ethanol droplet. Despite the glucamine modification of the surface resulting in a decrease in the static contact angle value of the water droplets from ~112° to ~88°, the water droplet does not spill on the tilted surface. Thus, the beaded water droplet failed to recognize the chemically modified track and responded differently than the beaded droplet of ethanol on the same patterned interface. Even a

large droplet of water (volume of 1 ml) placed across the pattern and non-patterned region displayed effortless sliding without leaving any trace of it on the patterned interface that kept tilted at 5° as shown in Supplementary Fig. 27. Hence, the current study revalidated that the patterned interface behaved differently for low and high surface tension liquids, even at low tilting angles. This principle would be capable of separating and collecting liquids at distinct locations just based on the difference in their $\gamma_{LV}$.

Thereafter, another wettability pattern was configured with an array of glucamine-modified spots on the omniphobic solid slippery interface to selectively capture one liquid from a mixture of two immiscible liquids (e.g., crude/water and hexanol/water), as shown in Fig. 6d. To demonstrate the separation of immiscible liquids of high and low surface tension from their mixture, three different mixtures of crude oil and water were placed on the patterned interface, where the content of crude oil was varied from 25% to 7%. During the translation of these liquid mixtures on the pattern interface, the chemically modified spots readily and selectively captured the crude oil and allowed the oil-free water droplet to travel away from the capture oil phase on the pattern interface (Fig. 6e and Supplementary Movie 3). With increasing the content of crude oil in the mixture, the traces of crude oil were observed to be more in the pattern interface and allowed the qualitative identification of the oil content in the oil/water mixture (Fig. 6e). Further, NMR characterization of both oil/water mixture and separated water phase was performed as shown in Fig. 6f, where the absence of characteristics $^1$H NMR peak of crude oil at 0.84 ppm and 1.24 ppm confirmed the successful separation of the collected water phase at the end of the pattern interface from crude oil. Thus, the prepared pattern interface acts as an oil/oily liquid-catcher. This system similarly performs for the separation of other water/immiscible organic solvents such as hexanol (Supplementary Fig. 28).

Further, such a patterned interface was utilized to demonstrate the extraction of an organic compound from one liquid to another, enabling targeted delivery of the extracted compound to the specified location (Fig. 6g). In this context, a droplet (5 μl) of hexanol was placed on the patterned interface. Thereafter, a 60 μl droplet of rhodamine B (RhB) dyed aqueous solution was introduced atop the hexanol droplet, causing the cloaking of hexanol layer around the aqueous droplet (Fig. 6h). Over time, RhB started to diffuse from water to hexanol. Approximately after 15 min, RhB mostly transferred from water to hexanol, leaving a colourless water droplet and the separation was visible to the naked eye (Fig. 6h). Subsequently, on tilting the pattern interface, the dyed hexanol droplet was moved around and was captured by the glucamine-modified spots, allowing the separated water droplet to slide down the interface (Fig. 6h). Further, UV-Vis absorbance of the aqueous droplet was measured at 550 nm before and after separation to characterize the transfer of RhB from the aqueous phase to hexanol, a nearly negligible amount of RhB was found in water droplet after the detachment and separation of

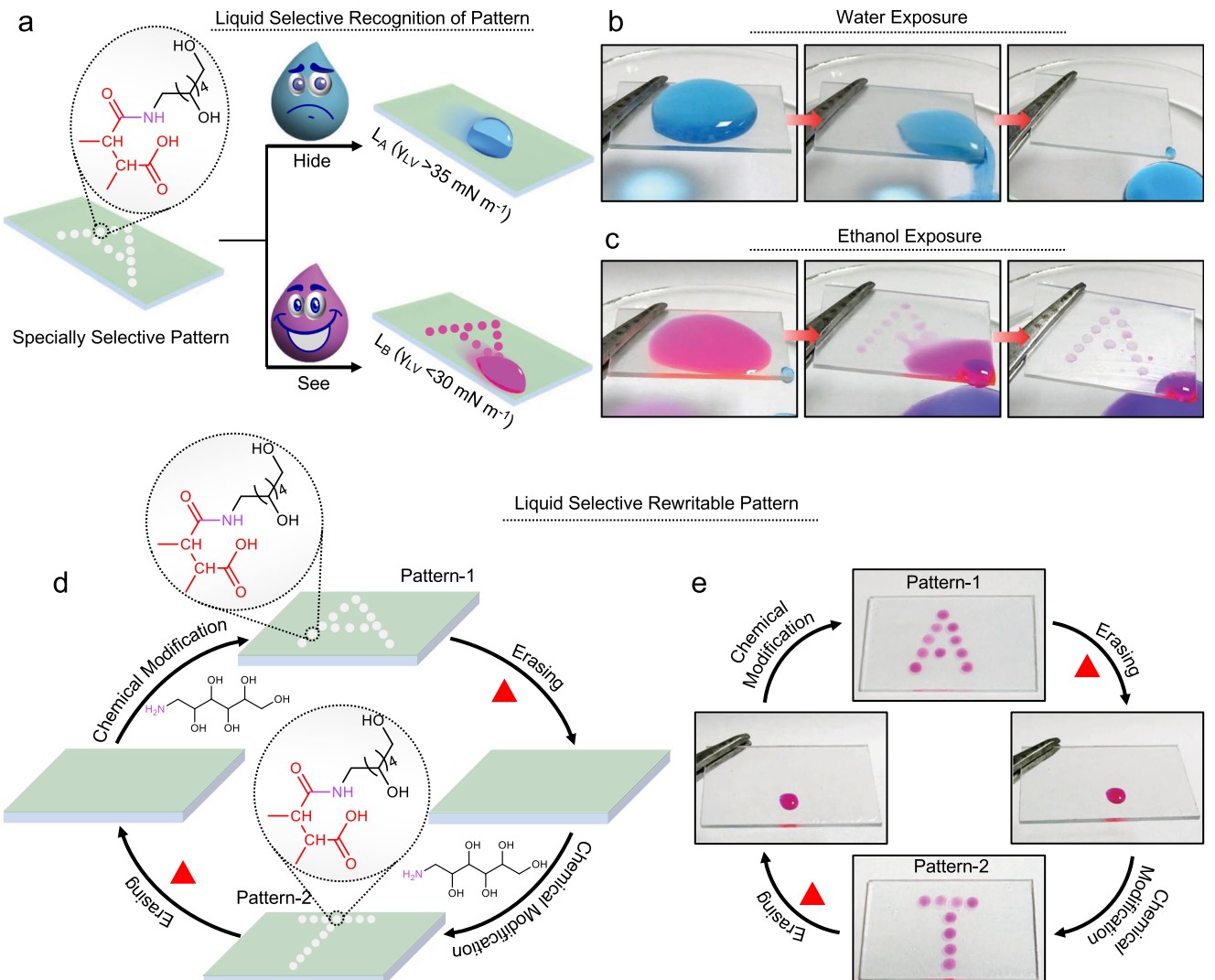

**Fig. 5 | Rewritable and 'liquid specific' recognizable pattern. a** Schematic illustrating the liquid-specific recognition of glucamine-modified spatially selective pattern (in the shape of 'A') by only low surface tension liquids (< 30 mN m⁻¹), whereas the pattern remain unrecognizable by high surface tension liquids (> 35 mN m⁻¹). **b** Photographs showing the inability of water droplets to recognize the glucamine-modified spatially selective pattern (in the shape of "A").

**c** Photographs showing the recognition of spatially selective pattern (in the shape of "A") during the exposure of ethanol droplet. **d**–**e** Schematic (**d**) and photographs (**e**) demonstrating the rewritability of spatially selective patterns in different architectures (in the shape of "A" and "T") through chemical modification with glucamine and subsequent heat treatment to erase the pattern.

from hexanol (Fig. 6i), indicating successful transfer and delivery of RhB from water to patterned spots thorough liquid-liquid extraction process on the patterned surface.

Thereafter, the pattern wettability on omniphobic solid slippery interface was strategically utilized to demonstrate its potential application to prepare polymer film with different shapes through interfacial polymerization, where the wettability pattern interface acts as a reusable template. Firstly, glucamine-modified lotus-shaped pattern was prepared on the slippery interface. Then, an ethanolic solution of polyethylene glycol diacrylate (PEGDA) and photoinitiator 2-hydroxy 2-methyl propiophenone was introduced to the patterned interface to selectively capture the ethanolic solution of monomer only on the lotus-shaped pattern (Fig. 7b, c). Then, this surface was irradiated with UV light (365 nm) for 5 min to initiate the photopolymerization reaction (Fig. 7a). ATR-FTIR was performed to characterize the polymerization reaction of PEGDA (Fig. 7a), where 1410 cm⁻¹ peak corresponding to the C−H deformation of vinylic group was depleted completely after the polymerization reaction (Fig. 7d). Thereafter, a free-standing lotus-shaped polymer film of PEGDA (Fig. 7c) was collected by easily delaminating it from

the surface upon heating for about 5 min. Simultaneously, thermal treatment enabled the erasing of the lotus pattern from the surface. Further, the same procedure of surface polymerization was repeated by subjecting the same surface to create another spatially selective chemical modification with a different shape (ring). A self-standing polymer film of PEGDA in a ring shape was collected and delaminated from the pattern interface (Fig. 7c). The same interface can be successfully applied to repeat this process for at least 20 cycles. Thus, this method could be useful for different interfacial polymerization processes, eliminating the requirement for photomasks of different shapes and the wastage of chemicals. Recently, reports of high throughput liquid-liquid extraction on miniaturized platforms have gained major attention due to its significance in the synthesis of compounds or relevant drug molecules on miniaturized patterned regions, chemical characterization on a chip, biological screening, environmental remediation, etc[17,53–56]. The current design of chemically reactive omniphobic solid slippery interface having the ability to incur and (or) erase chemical modifications can be further strategically combined with microchannel cantilever based contact printing method[57–59] to design rewritable and smart liquid

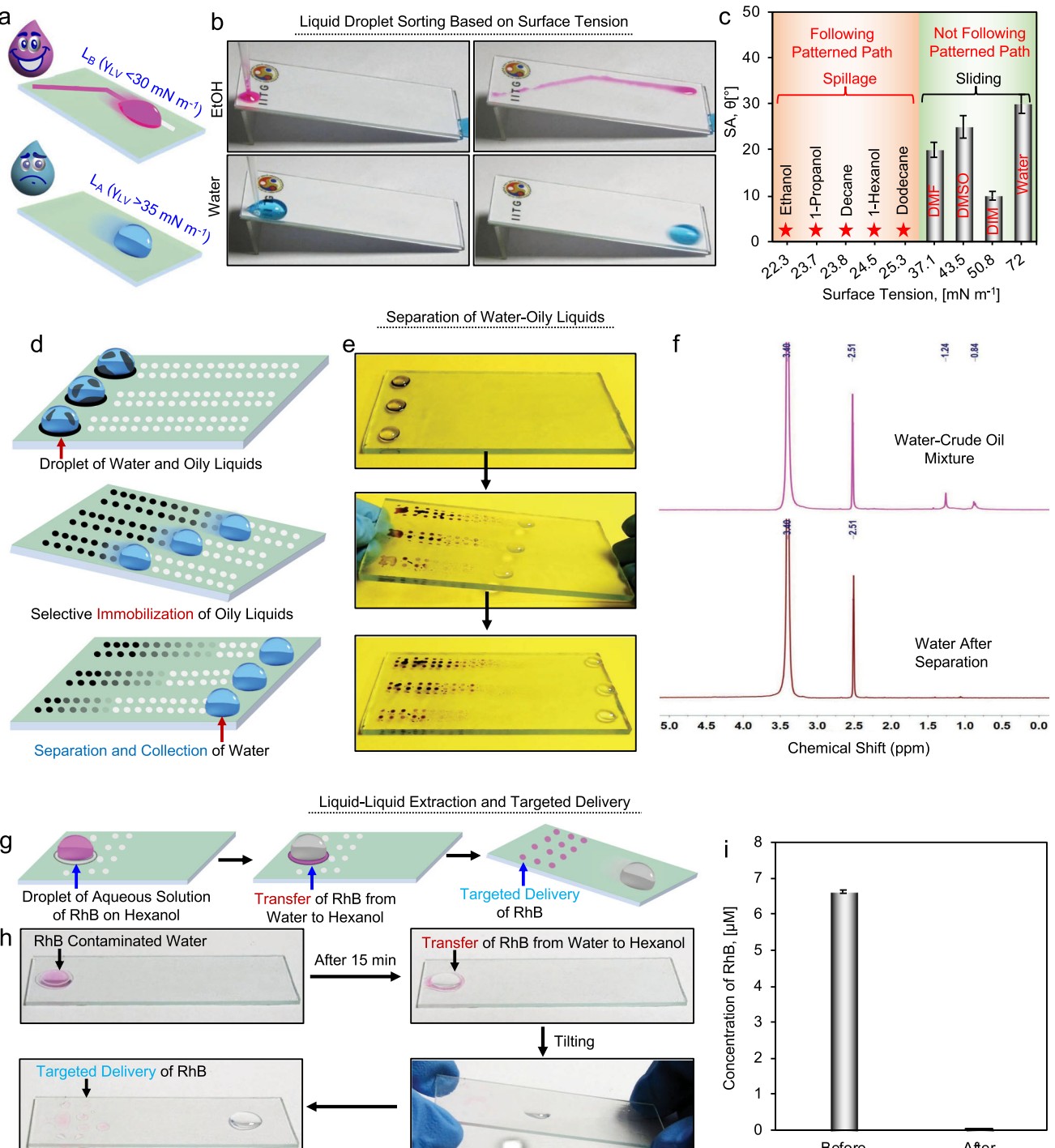

**Fig. 6 | Pattern wettability for screening and separating liquids. a** Schematic illustrating the guided transport of low surface tension liquids on the glucamine-modified (denoted as glu-modified) interface, whereas the pattern remains unrecognizable by high surface tension liquids, therefore high surface tension liquids followed a distinct path on same tilted interface. **b** Photographs showing the guided transport of low surface tension liquids on the pattern interface, whereas high surface tension liquids failed to recognize the pattern—and followed a different path on the same interface to slide down. **c** Plot accounting for the change in sliding angle (SA) for different liquids with a wide range of surface tensions on glu-modified surfaces. Orange and green shades indicate the ability of beaded liquids for following and not following the patterned path, respectively. **d, e** Schematic and photographs demonstrating the separation of water-crude oil droplets mixture on the patterned interface, where glu-modified patterned region selectively collected the oil droplets—and allowed water droplet to slide effortlessly. **f** NMR spectra of crude oil-water mixture before and after sliding on the pattern interface. **g, h** schematic (**g**) and photographs (**h**) demonstrating the extraction of rhodamine B (RhB) from the aqueous phase to hexanol followed by targeted delivery to the specified patterned region of the prepared interface. **i** Plot accounting the RhB concentration in the aqueous phase before and after liquid-liquid extraction on the pattern interface. The error bar indicates the standard deviation with number of measurements, $n = 3$ for each data point. Source data are provided as a Source Data file.

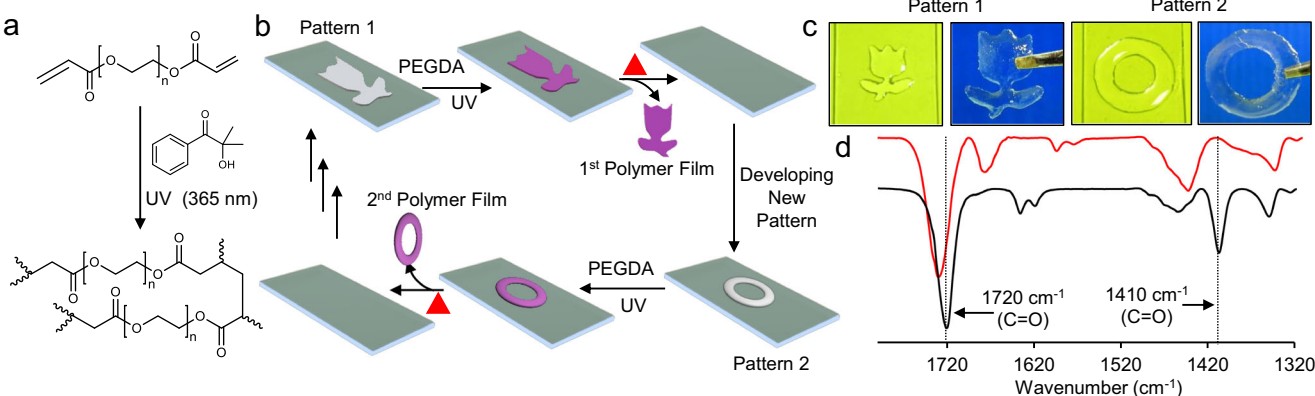

**Fig. 7 | Moulding of non-aqueous liquids on surface. a** Schematic of photopolymerization reaction of polyethylene glycol diacrylate (PEGDA). **b** Schematic illustrating the utilization of the surface and patterned wettability as a reusable template for spatially selective polymer synthesis on the pattern interface by confining ethanolic solution of PEGDA with desired shape. **c** Photographs of ethanolic solution of PEGDA taking the shape of different patterns (in the shape of lotus and ring) and corresponding polymeric film obtained through interfacial polymerization. **d** FTIR spectra before (black) and after (red) photopolymerization of PEGDA. Source data are provided as a Source Data file.

selective miniaturized pattern for expanding the scope of liquid-liquid extraction, miniaturized chemical synthesis, etc.

## Discussion

In summary, we have successfully and unprecedentedly introduced a facile approach to develop a chemically reactive and healable omniphobic solid slippery interface by infusing a crystalline copolymer PODAMA in a porous matrix. In the current design, the gradual alteration in the content of MA moiety in PODAMA allows the modulation of two important parameters i.e., roughness and SFE of the prepared coating. The surface roughness of prepared coatings was gradually decreased from ~48 ± 3.7 nm to ~0.52 ± 0.1 nm, as the content of MA moiety in the crystalized network of polymer increased. However, the coating with the lowest roughness failed to slide low surface tension liquids ($< 30 \, mN \, m^{-1}$). Interestingly, the SFE of these prepared coatings gradually increased from ~20 ± 0.8 $mN \, m^{-1}$ to ~32 ± 0.2 $mN \, m^{-1}$ on increasing the content of MA moiety. The crystalized network of polymer continued to display antiwetting property against low surface tension liquid only and when the SFE of the coating remains $<25 \, mN \, m^{-1}$. While co-optimization of surface roughness and SFE is essential to achieve omniphobic solid slippery property, here the SFE played a superior role over roughness to confer slippery property towards low surface tension liquids. Further, a strategic chemical functionalization of the amine-reactive MA group present in the crystalline omniphobic solid slippery with hydrophilic glucamine molecule provided a facile basis to moderately elevate the SFE. As a result, the chemically modified interface failed to slide droplets of low surface tension liquids ($<30 \, mN \, m^{-1}$)—rather spilled on the tilted interface. Whereas high surface tension liquids still can slide on the same surface. Eventually, such a strategy allowed a 'liquid specific' sacrifice of embedded antiwetting property of the omniphobic solid slippery interface. Further, the phase–transition of the crystalline polymer allowed it to undo the entire process and the chemically modified coating regained its native omniphobic solid slippery property. The rearrangement of the crystalline polymeric network allowed to hide hydrophilic domain and exposed loaded MA moiety and allowed further post-modification of such interface. But such a process can be successfully repeated for limited cycles. This concept was further utilized for spatially selective modification to develop liquid-specific rewritable patterns with limited cycles of erasing and rewriting ability on the same surface. In this current study, the rewritable pattern on slippery coating is derived through a stable covalent reaction between glucamine and available MA moiety of the copolymer—and thus the limited availability of MA moiety in the prepared coating allowed rewriting the pattern for a certain number of cycles. To address this issue, spatially

selective chemical modification may be attempted following a dynamic covalent bonding as an alternative approach. So that the chemical modification can be erased in the presence of specific and appropriate stimuli, such as pH, temperature, UV light, etc. to recover the native functional group, and the same interface can be reutilized for rewriting the pattern. For example, an essential chemical modification can be achieved through associating imine bond—which is known to be labile towards acidic hydrolysis[60]. Such an approach would allow the pattern to be rewritten for several cycles as the same reactive moiety is reversibly consumed and subsequently reused to alter the surface free energy. Such strategy will be explored separately in future studies. Moreover, the patterned interface was successfully applied to demonstrate screening liquids based on their surface tension, separation of immiscible liquid mixtures and moulding of liquids for interfacial polymerization. Such unique ability of the developed chemically reactive omniphobic solid slippery interface can be combined with state of art molecular printing approach to derive various functional and miniaturized patterned interfaces that would have potential applications in microfluidics devices, biofluidic separation, cell culture, high throughput sensing, biomedical engineering, and so on.

## Methods
### Materials
Dipentaerythritol pentaacrylate (5-Acl; molecular weight: 524.51 g mol⁻¹), branched polyethylenimine (BPEI; Mw ~800 Da), octadecyl acrylate (ODAc; purity: 97%; CAUTION: causes skin, eye and respiratory irritation), lauryl acrylate (LAc; purity: 90%; CAUTION: causes skin, eye and respiratory irritation), polyethylene glycol diacrylate (PEGDA; CAUTION: causes skin irritation and serious eye damage), 2-hydroxy-2-methylpropiophenone (purity: 97%), methylene blue (purity: ≥82%), rhodamine B (RhB; purity: ≥95%), Nile red (purity: ≥97%), decane (purity: ≥99%), dodecane (purity: ≥99%), diiodomethane (DIM; purity: ≥98%), were purchased from Sigma Aldrich (Bangalore, India). Docosyl acrylate (DAc; purity: >95%; CAUTION: causes skin and serious eye irritation), maleic anhydride (MA; purity: >99%), D-glucamine (purity: >97%) were purchased from Tokyo Chemical Industry (India) Pvt. Ltd. Azobisisobutyronitrile (AIBN; purity >98%) was purchased from Avra Synthesis Pvt. Ltd. Dimethyl formamide (DMF; purity: ≥99.9%), dimethyl sulfoxide (DMSO; purity: ≥99.9%), 1-propanol (purity: 99.5%; CAUTION: causes eye damage and flammable), 1-hexanol (purity >99%; CAUTION: causes eye irritation and flammable) were purchased from Alfa-Aesar, India. Ethanol was procured from Changshu Hongsheng Fine Chemical. Toluene was purchased from Finar, India. Crude Oil was

procured from Oil India Refinery, Assam. Diesel and petrol were obtained from the nearest Indian oil petrol pump in Guwahati (Assam, India). Kerosene was purchased from a local shop (Assam, India). Vegetable oil, motor oil and spray bottle was purchased from Amazon India. Microscopic glass slides were purchased from JSGW (Jain Scientific Glass Works), India.

## Characterisations

Static contact angles and sliding angles were recorded at ambient conditions using the KRUSS Drop Shape Analyzer-DSA 100E instrument, where the static contact angle measurements were done with 5 μL-sized water droplets at three different positions of each sample. Surface free energy of the coated interface was determined by static contact angle measurements of two liquids including non-polar diiodomethane and polar distilled water using Kruss Drop Shape Analyzer- DSA100. Liquid needle dosing system DS3252 was used to dispense liquid droplets on the surface and static contact angles of these liquid droplets were analysed with the ellipse fitting method in the software version 1.16.0. Owens, Wendt, Rabel and Kaelble (OWRK) method was followed to calculate the surface free energy with the contact angle data of these two liquids. All measurements were conducted at ambient conditions, with three distinct measurements recorded for each coating. Atomic force microscope (AFM) images were recorded using OXFORD Cypher Atomic Force Microscope. Carl Zeiss field emission scanning electron microscope (FESEM) was used to capture scanning electron microscope images for surface morphology analysis. Samples were sputtered with a thin layer of gold for 120 S before capturing electron microscope images. PerkinElmer UTAR Two was used to record attenuated total reflection-fourier transform infrared (ATR-FTIR) spectra at ambient conditions. The X-ray photoelectron spectroscopy (XPS) spectra were recorded in PHI5000 VersaProbe III (Φ ULVAC-PHI, INC.) equipped with a monochromated Al Kα X-ray source. NMR spectra were recorded using Bruker 400 MHz NMR spectrometer. The UV absorbance data was measured in Thermo scientific multiskan G.O. Digital images were captured using a Canon powershot SX540HS camera. The thickness of the coating was analysed using Stylus surface profilometer, Veeco (Dektak 150). Milli-Q grade water was used for all the experiments.

## Preparation of reactive crystalline comb-like random copolymers varying the content of reactive moiety

The random copolymer, polyoctadecyl acrylate-co-maleic anhydride (PODAMA) was prepared following the solution radical polymerization technique. ODAc (6 mmol, 2 g) and MA (3 mmol, 0.3 g) were added into 20 ml of toluene in a round bottom flask and sonicated to form a homogeneous mixture. This prepared solution of monomer was then purged with nitrogen before the addition of initiator AIBN (0.12 mmol, 20 mg). Subsequently, the reaction mixture was refluxed at 80 °C under an $N_2$ atmosphere for 6 h. Then, the reaction mixture was brought down to room temperature and the product was isolated from the solution using the solvent exchange method by the addition of 150 ml cold ethanol. The product was precipitated out as a white solid and collected by centrifugation. For further purification, the collected solid polymer was dissolved in 5 ml THF and phase separated by adding 50 ml cold ethanol. This process was repeated 3 times and finally, the obtained product was dried at 70 °C for 6 h, yielding 1 g (43%) of isolated product, which is denoted as $PODAMA_2$. The same above-mentioned procedure was followed to prepare PODAMA with other compositions, where amounts of MA were varied while maintaining similar amounts of ODAc, as used in the preparation of $PODAMA_2$. Specifically, 0.2 g, 0.6 g, and 1.2 g of MA were used to prepare different other polymers denoted as $PODAMA_1$, $PODAMA_3$, and $PODAMA_4$, respectively. The yields of obtained polymers are 1.2 g (54%) for $PODAMA_1$, 1 g (38%) for $PODAMA_3$ and 1 g (31%) for $PODAMA_4$. The prepared polymers were characterized by FTIR and NMR spectroscopic analysis.

## Preparation of comb-like copolymers with varying the lengths of hydrocarbon tail

In addition to PODAMA, two different random copolymers—i.e., poly-lauryl acrylate-co-maleic anhydride (PLAMA) and polydocosyl acrylate-co-maleic anhydride (PDAMA) were prepared to have the variation in the hydrocarbon chain length of the alkyl acrylate moiety. The PLAMA was prepared following the same procedure mentioned above following the solution radical polymerization technique. LAc (6 mmol, 1.4 g) and MA (3 mmol, 0.3 g) were added into 20 ml of toluene in a round bottom flask and sonicated to form a homogeneous mixture. This monomer solution was then purged with nitrogen before the addition of initiator AIBN (0.12 mmol, 20 mg). Subsequently, the reaction mixture was refluxed at 80 °C under an $N_2$ atmosphere for 6 h. Then, the reaction mixture was brought down to room temperature and the product was isolated from the solution using the solvent exchange method by the addition of 80 ml cold methanol. The product was separated as a viscous colourless liquid and collected by centrifugation. For further purification, the collected polymer was again dissolved in 5 ml THF and phase separated by adding 50 ml cold methanol. This process was repeated 3 times. This process was repeated 3 times and finally, the obtained product was dried at 70 °C for 6 h, yielding 0.7 g (41%) of isolated product. The prepared polymer was characterized by FTIR spectroscopy.

Similarly, the PDAMA was prepared, where DAc (6 mmol, 2.3 g) and MA (3 mmol, 0.3 g) were added into 20 ml of toluene in a round bottom flask and sonicated to form a homogeneous mixture. This monomer solution was then purged with nitrogen before the addition of initiator AIBN (0.12 mmol, 20 mg). Subsequently, the reaction mixture was refluxed at 80 °C under an $N_2$ atmosphere for 6 h. Then, the reaction mixture was brought down to room temperature and the product was isolated from the solution using the solvent exchange method by the addition of 150 ml cold ethanol. The product was separated as white solid powder and collected by centrifugation. For further purification, the collected polymer was again dissolved in 5 ml THF and phase separated by adding 50 ml ethanol. This process was repeated 3 times and finally, the obtained product was dried at 70 °C for 6 h, yielding 0.5 g (27%) of isolated product. The prepared polymers were characterized by FTIR spectroscopy.

## Fabrication of omniphobic solid slippery coating

Omniphobic solid slippery coating was prepared by infusing PODAMA into a hydrophilic porous matrix. The hydrophilic porous matrix was prepared following a spray deposition technique[48]. First, two separate solutions of 5-Acl (1.325 g ml$^{-1}$) and BPEI (0.7 g ml$^{-1}$) were prepared in hexanol. Then, a reaction mixture solution prepared by mixing 10 ml solution of 5-Acl and 3 ml solution of BPEI was spray deposited manually from a 10 cm distance over an area of 200 cm$^2$. Then, the spray deposited substrate was kept in open air for 10 min and immediately washed with THF followed by drying at 70 °C for 5 min.

The next step involves the infusion of $PODAMA_2$ into the porous matrix. A 2 ml solution of PODAMA in THF (0.5 mg ml$^{-1}$) was infused in the prepared porous matrix with the dimension of 7.5 cm × 2.5 cm and kept at 70 °C for 15 min for complete evaporation of THF and subsequent melting of infused $PODAMA_2$. Upon cooling down to room temperature, the infused $PODAMA_2$ polymer again solidified, resulting in the formation of a solid slippery coating $RPIC_2$. For other prepared polymers, the same above-mentioned procedure was followed to prepare the respective polymer-infused coating.

## Liquid selective reversible transition of slippery property

The $RPIC_2$ was subjected to post-modification for the liquid selective reversible transition of antiwetting property. The coating was dipped horizontally in a DMSO solution of D-glucamine (5 mg ml$^{-1}$) and kept for 12 h. Then, the coating was thoroughly washed with DI water for

three times and kept in open air for about 1 h. Thereafter, the anti-wetting property of this modified surface was examined with different liquids having a wide range of surface tension.

Next, this post-modified coating was heated at 70 °C for 15 min and allowed to cool at room temperature to restore the native anti-wetting property.

## Development of patterned interface

The rewritable pattern on PODAMA$_2$ infused coating was developed by spatially selective modification of the interface using D-glucamine solution (5 mg ml$^{-1}$) in DMSO. Tiny droplets (2 μl) of D-glucamine in DMSO were placed on PODAMA$_2$ infused surface in a particular arrangement and kept undisturbed for 12 h to get the desired architecture. Then, the surface was washed with DI water three times.

## Development of pattern for liquid sorting

A continuous channel was made by placing a paper soaked in glucamine solution in DMSO on the PODAMA$_2$ infused coated surface for 12 h. Then, the surface was washed with DI water three times. Following this, different liquids were introduced onto this patterned coating and the movement of the liquids was examined on a tilted interface.

## Polymerization by liquid molding on pattern surface

Different patterns for interfacial polymerization were developed by placing paper soaked in glucamine solution in DMSO, shaped as either lotus or ring on the surface for 12 h. Then, the surface was washed with DI water three times. Thereafter, the ethanolic solution of PEGDA (20 mg ml$^{-1}$) and photoinitiator 2-hydroxy 2-methyl propiophenone was introduced on the patterned region and kept under UV for photopolymerization. Next, the polymer film was delaminated from the surface by simply heating the interface at 70 °C for 15 min and again reused to prepare another pattern following the same procedure.

## Data availability

The data supporting the findings of the study are included in the main text and supplementary information files. All raw data can be obtained from the corresponding author upon request. Source data are provided with this paper.

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

## Acknowledgements

The financial supports from Science and Engineering Research Board (CRG/2022/000710), Ministry of Electronics and Information Technology (no. 5(1)/2022-NANO), DST (DST-FIST programme; Sanction No. SR/FST/CS-II/2017/23 C) and DBT (BT/PR45283/NER/95/1919/2022) are acknowledged. U.M. thanks CIF, CFN, SHST and the Department of Chemistry, Indian Institute of Technology Guwahati, for their generous assistance in executing various experiments and for the infrastructures. M.D. and D. S. thank the MoE, India and UGC for their doctoral fellowships.

## Author contributions

MD designed, performed experiments, and carried out the analysis. DS performed the experiments and analysed the data. SAR and DG contributed to perform experiments. AD helped in designing experiments and analysing data. UM conceived the idea, supervised the work, and wrote the manuscript, and all authors contributed to editing and reviewing the manuscript.

## Competing interests

The authors declare no competing interests.
