## [Peer Review File · Nature Communications]

'Rewritable' and 'Liquid-specific' recognizable wettability patternREVIEWER COMMENTS

Reviewer #1 (Remarks to the Author):

I have considered the manuscript by Dhar et al. entitled 'Rewritable' and 'Liquid-specific' recognizable wettability pattern. In this article the authors take advantage of a crystalline network of phase-transitioning polymer (PODAMA), which is chemically reactive and can go heating-cooling cycle to "rejuvenate" the surface.

The central premise of the article is interesting, and indeed the possibility of realising rewritable and liquid-specific wettability pattern is exciting for the field of wetting and surface chemistry. The authors have also employed a number of techniques to better understand their experimental results. For sure this work deserves to be published, however I am not sure it is suitable for Nature Communications, at least in its current form. My reservations are below.

1. To me, the conclusions of the work seem to be a bit of an overclaim for three reasons.

a) If I investigate their results, Fig. 2 for example, the CAH (contact angle hysteresis) and sliding angle are not that small, especially compared to other types of liquid repellent surfaces (e.g. SLIPS, SOCAL, and some superhydrophobic surfaces).

b) Indeed, there are differences in CAH (which is not surprising), and this is what the authors have exploited to have some liquid droplets sliding and some pinned. However, I suspect this intrinsically depends on what tilt angle is used. Too small and no droplets move (not selective). Too large and many liquids slide (again, not selective). From the paper, it is unclear how big/small the range of tilt angle when some selective sliding can be realised. In real applications, the tilt angle cannot always be optimised.

c) This work relies on a particular polymer (PODAMA) and mainly look into water vs ethanol droplet sliding. In general, real applications necessitate the use of different surface chemistry and different liquids. From the work presented, it is unclear what physical/chemical principles one may generalise to be able to use other materials.

2. The ability to "rewrite" the patterns is interesting. The authors claim that the Glu domains become buried upon heating-cooling cycle and fresh reactive MA moiety is seen at the surface. Is there a limit in terms of the number of cycles until the system becomes saturated, beyond ~ 20 cycles that the authors have done?

3. The authors claim that some of the surfaces (RIPC4) have lower roughness and yet have larger friction force for sliding. This suggests the CAH is higher. What is the origin of the CAH? Are there variations in the surface energies that can lead to pinning points?

4. Related to points 1b and 3) above, when the authors claim that water droplets do not see the patterns, what happens at small tilt angle? Is there effectively a chemical domain (even a diffuse one) that leads to droplet pinning between regions with and without chemical modifications? Based on the presented work, I expect chemical domains with CA 88 and 112 degrees, which the droplet will see when the gravitational force is not large.

Reviewer #2 (Remarks to the Author):

Dhar and colleagues present a clever approach via selective modification of a (semi)crystalline polymer network such that during a thermally-mediated phase transition, the network reverts to its native wettability. The authors present a compelling narrative for reversible wettability, but the process is not quite fully reversible. After treating the surface with glucamine, a ring opening reaction is triggered with the maleic anhydride monomers that have co-polymerized onto the polymer chains next to neighboring, varying length n-alkyl side chains off of the acrylates in the chain backbone. While heating and cooling the material does "erase" the crystallinity, it does not seem that the glucamine is removed from the polymer network, rather the side chain semi-crystallinity is disrupted. Over repeat cycles, the accessible maleic anhydride will become less and less leading to a system that is no longer able to be further surface modified due to using up the available MA in the network.

The authors claim rewrite-ability over "multiple" cycles, which is likely largely dependent on the relative ratio of Glu to MA (or the ink density per rewrite cycle). The authors would do well to bound these statements if not experimentally at least with a theoretical discussion of the limits of rewrite-ability, which are likely related to this ink density metric per writing cycle. The authors could also perhaps imagine or suggest future work where a thermally labile reaction could replace the covalent Glu-MA reaction and perhaps form a more reusable surface.

Nonetheless, these critiques should not diminish from an excellent story, compelling videos and well-engineered figures both within the main text and in the supplementary information provided.

The work is detailed and likely reproducible in a well-maintained chemistry or polymer science laboratory. I applaud the authors for interesting and well-documented work that could lead to other interesting avenues in the field in the years to come.

Point-by-point response to the reviewer's comment

Responses to Reviewer 1:

I have considered the manuscript by Dhar et al. entitled 'Rewritable' and 'Liquid-specific' recognizable wettability pattern. In this article the authors take advantage of a crystalline network of phase-transitioning polymer (PODAMA), which is chemically reactive and can go heating-cooling cycle to "rejuvenate" the surface.

The central premise of the article is interesting, and indeed the possibility of realising rewritable and liquid-specific wettability pattern is exciting for the field of wetting and surface chemistry. The authors have also employed a number of techniques to better understand their experimental results. For sure this work deserves to be published, however I am not sure it is suitable for Nature Communications, at least in its current form. My reservations are below.

Response: We are grateful for the reviewer's positive evaluation of our manuscript and thank the reviewer for giving us the chance to revise our manuscript. In light of the referee's comments, we have significantly addressed these specific points as you suggested. Our point-by-point answers are listed below.

1. To me, the conclusions of the work seem to be a bit of an overclaim for three reasons.

a) If I investigate their results, Fig. 2 for example, the CAH (contact angle hysteresis) and sliding angle are not that small, especially compared to other types of liquid repellent surfaces (e.g. SLIPS, SOCAL, and some superhydrophobic surfaces).

Response: We thank the reviewer for raising this concern. It is true that the sliding angle of the currently synthesized absolutely solid slippery interface is relatively higher in comparison to slippery liquid infused porous surface (SLIPS), and other quasi-solid slippery interface, as these interfaces are predominantly prepared either completely or partially infusing liquid lubricants or by covalent immobilization of liquid lubricants. Mostly such interface remained wet and suffered from unavoidable loss of lubricants. Whereas, superhydrophobic interface is an example of heterogeneous wettability where contact area between beaded liquid and solid surface is minimal and the sliding angle is low, however, such interface is less tolerant to physical abrasions due to presence of metastable trapped air in hierarchically rough/porous interface decorated with low surface energy coating. However, in comparison to the recent reports on solid slippery interfaces, the sliding angle of beaded liquid is highly comparable as depicted in **Table R1**.

Methodology	SA (°) of Water	SA (°) of Ethanol	References
Paraffin wax infusion	90°	-----	J. Mater. Chem. A 9 , 16974–16981 (2021)
Paraffin wax infusion	33.2 ± 2.8°	-----	J. Mater. Chem. A 6 , 16355–16360 (2018)

Surface growth of MOF	15°	15°	Nano Lett. 21 , 3480–3486 (2021)
Paraffin wax infusion	27°		ACS Nano 14 , 10198–10209 (2020)
Solvent-free blade method	24±2°	12±1°	Adv. Mater. 34 , 2108232 (2022)
In-situ deposition of hydrophobic nano-complexes	15°	10°	Chemical Engineering Journal 465 , 142776 (2023)
Reactive and crystalline polymer infusion	20°±1.6°	10°±1°	This Work

Table R1. Comparison table based on sliding angle of beaded liquid of this work with the recent reports on solid slippery interfaces

We have incorporated the following statement in the manuscript

“ The sliding angle of beaded liquid droplets of the developed surface is highly comparable with the recent reports on solid slippery interfaces, as depicted in Table R1.”

In addition, we would like to emphasise that the prepared solid slippery interface remained chemically reactive and enabled healing of both physical damages and chemical modifications/perturbation on such interface for multiple cycles—and the report of such material is unprecedented in the relevant literature.

b) Indeed, there are differences in CAH (which is not surprising), and this is what the authors have exploited to have some liquid droplets sliding and some pinned. However, I suspect this intrinsically depends on what tilt angle is used. Too small and no droplets move (not selective). Too large and many liquids slide (again, not selective). From the paper, it is unclear how big/small the range of tilt angle when some selective sliding can be realised. In real applications, the tilt angle cannot always be optimised.

Response: We thank the Reviewer 1 for pointing out this concern. However, the statement reflected a slight misunderstanding of the current work. *In the present work, we have not exploited droplet sliding versus droplet pinning behaviour of the prepared slippery coating—rather, it is a demonstration of surface tension specific liquid droplet sliding versus droplet spillage phenomena.* The optimized chemically reactive slippery coating, denoted as RPIC₂ exhibits sliding property towards various polar and non-polar solvents having surface tension ranging from 22 mN/m to 72 mN/m as shown in Supplementary Fig. 8. Further, subsequent modification of RPIC₂ with primary amine group containing hydrophilic small molecule, i.e., glucamine, resulted in the selective compromise of sliding property of liquid droplets with surface tension < 30 mN/m. *Primarily, glucamine modification resulted in a coating (Glu-modified RPIC₂) which allowed arbitrary and instant spillage of beaded liquids having surface tension < 30 mN/m (Supplementary Fig. 16), whereas, on the contrary, droplets of liquids having surface tension > 35 mN/m continued to slide on the same Glu-modified RPIC₂ (Supplementary Fig. 17).* For example, beaded droplet (10 µl) of decane, dodecane, ethanol, 1-

propanol and 1-hexanol displayed effortless sliding on tilted (15°) interface of RPIC₂ as shown in Supplementary Fig. 16a. However, after chemical modification of the same interface with glucamine, beaded droplets of decane, dodecane, ethanol, 1-propanol and 1-hexanol failed to slide, rather the beaded droplets of these liquids arbitrarily spilled on the Glu-modified RPIC₂ (Supplementary Fig. 16b).

Supplementary Fig. 16. a Digital images depicted the sliding of beaded droplets (volume $10\ \mu\text{l}$) of liquids having surface tension (γ) $< 30\ \text{mN/m}$ on tilted (15°) interface of reactive polymer infused coating-2 (RPIC₂). **b** Digital images depicted the arbitrary spillage of beaded droplets (volume $10\ \mu\text{l}$) of liquids having surface tension (γ) $< 30\ \text{mN/m}$ (volume $10\ \mu\text{l}$) on tilted (15°) interface of glucamine modified RPIC₂.

Thus, such chemical modification based alteration of wetting behaviour from sliding to spillage was only observed for beaded liquid droplets having surface tension $< 30\ \text{mN/m}$ (Supplementary Fig. 16, 18). Moreover, such selective spillage of liquid droplets on the modified interface was observed over a wide range of tilting angles (Supplementary Fig. 19) and beaded droplets of liquids having surface tension $< 30\ \text{mN/m}$ were found to be spread down the slope, irrespective of the volume of selected liquids having surface tension $< 30\ \text{mN/m}$ ((Supplementary Fig. 20).

Supplementary Fig. 17. a-b) Digital images depicted the sliding of beaded droplets of liquids having surface tension (γ) > 35 mN/m on tilted (20°) interface of RPIC₂ before (a) and after (b) glucamine modification.

Supplementary Fig.19. Digital images illustrating the spillage of liquid droplets (volume 10 μ l) having surface tension (γ) < 30 mN/m on glucamine modified RPIC₂ over a range of tilting angles from 5° to 30°.

Supplementary Fig. 20. Digital images illustrating the spillage of liquid droplets of different volumes (3 μ l, 10 μ l, 25 μ l and 50 μ l) having surface tension (γ) < 30 mN/m on the tilted interface (5°) of glucamine modified RPIC₂.

Further, we do agree with the reviewer’s comment that tilting angle cannot always be optimized in real world applications. *However, based on the series of additional experiments discussed above, we have demonstrated that the modified RPIC₂ displayed selective spillage of beaded liquids of surface tension < 30 mN/m, independent of the tilting angle of the modified coating and the volume of beaded liquids.*

We have incorporated the following statement in the manuscript:

“In fact, after chemical modification of RPIC₂ with glucamine, beaded droplets (10 μ l) of other liquids of low (< 30 mN/m) surface tension, including decane, dodecane, ethanol, 1-propanol and 1-hexanol failed to slide, instead the beaded droplets of these liquids arbitrarily spilled on the Glu-modified RPIC₂ (Supplementary Fig. 16b). Whereas, on the contrary, droplets of other liquids having surface tension > 35 mN/m continued to slide on the same modified interface (Supplementary Fig. 17b). However, prior to its post covalent modification, droplets of both low (< 30 mN/m) and high surface tension liquids effortlessly slide on the tilted interface of RPIC₂ as shown in Supplementary Fig. 16a and 17a. Thus, such chemical modification-based alteration of wetting behaviour from sliding to spilling was only observed for beaded droplets

of liquids having surface tension < 30 mN/m (Supplementary Fig. 16-18). Moreover, such selective spilling of liquid droplets (surface tension < 30 mN/m) was observed over a wide range of tilting angles of the modified interface (Supplementary Fig. 19). Also, beaded droplets of liquids having surface tension < 30 mN/m were found to be spread down the slope, irrespective of the volume of selected liquids (Supplementary Fig. 20). Thus, the prepared interface would be appropriate for real-world applications, where the optimization of the tilting angle of surface is not required to monitor the selective spillage of beaded liquids of surface tension < 30 mN/m.”

c) This work relies on a particular polymer (PODAMA) and mainly look into water vs ethanol droplet sliding. In general, real applications necessitate the use of different surface chemistry and different liquids. From the work presented, it is unclear what physical/chemical principles one may generalise to be able to use other materials.

Response: We are extremely sorry for the confusion. In the current study, we have prepared different sets of co-polymer, i.e. polyoctadecylacrylate maleic anhydride (PODAMA), varying the composition (in mole fraction) of maleic anhydride (MA) and octadecyl (ODA) moieties. The prepared coating, RPIC₂ derived by infusing the polymer of a particular composition of PODAMA, i.e. PODAMA₂ into a porous matrix, exhibited sliding behaviour not only towards water and ethanol but also a wide range of polar and non-polar liquids having surface tension ranging from 22 mN/m to 72 mN/m (Fig. 2h and Supplementary Fig. 8). The strategic post-chemical modification with glucamine allowed for selective compromise of the sliding behaviour of beaded liquids (decane, dodecane, ethanol, 1-propanol, 1-hexanol) having surface tension < 30 mN/m—and such liquids readily spilled/spread on the tilted interface, irrespective of the tilting angle (Supplementary Fig. 19). Hence, the prepared interface is not only applicable to ethanol Vs water, but also suitable for other pairs of polar and non-polar liquids as well.

However, for broader practical utilization, we have further synthesized another co-polymer polydocosyl acrylate maleic anhydride (PDAMA) incorporating a higher analogue of octadecyl moiety, i.e. docosyl moiety. The coating derived from infusing PDAMA into a porous matrix (denoted as RPIC_{PDAMA}) enabled the sliding of commercially available refined oils (motor oil, vegetable oil, diesel, petrol, kerosene) and complex crude oil (Supplementary Fig. 10) in addition to polar and non-polar liquids. Actually, the increase in the hydrocarbon chain length of the selected co-polymer provided a surface free energy of ~ 20 mN/m, which is even lower than RPIC₂. However, after the post-covalent modification with glucamine, droplets of oils and organic solvents with low (< 30 mN/m) surface tension readily spilled on the tilted interface, as shown in Supplementary Fig. 21. Further study revealed that the surface free energy of the coating elevated from ~ 20 mN/m to ~ 35 mN/m because of glucamine modification. Such chemical modification merely perturbed the surface roughness of the coating, as shown in Supplementary Fig. 22. In addition to this, another selected copolymer (polylauryl acrylate maleic anhydride, PLAMA) having a short hydrocarbon chain (i.e. $n = 9$ (lauryl acrylate, LAc)) failed to repel both low and high surface tension liquids (Fig. 1d and Supplementary Fig. 6). Hence, mostly, the alteration of surface chemistry contributes to selectively compromise the sliding behaviour of low surface tension (< 30 mN/m) organic solvents and refined/crude oils.

Supplementary Fig. 21. Digital images depicted the arbitrary spillage of beaded droplets (volume 10 μl) of organic solvents having surface tension (γ) < 30 mN/m and oils on tilted interface of glucamine modified reactive polydocosyl acrylate maleic anhydride (PDAMA) infused coating (RPIC_{PDAMA}).

Supplementary Fig. 22. Plot accounting for the changes in surface free energy (SFE) and root mean square roughness (Rq) of unmodified and glucamine modified reactive polydocosyl acrylate maleic anhydride (PDAMA) infused coating (RPIC_{PDAMA}). The error bar indicates the standard deviation with number of measurements, n=3 for each data point.

This study reveals that the strategic modulation of chemistry provides a basis to efficiently alter the liquid sliding/spilling behaviour on a solid surface, and such principle of chemical modulation would allow to derive other functional materials in future.

We have incorporated the following statements in the manuscript;

“Similarly, another chemically reactive comb-like polymer having a higher analogue of alkyl chain, i.e. polydocosyl acrylate maleic anhydride (PDAMA) was infused in a porous coating to derive solid slippery coating (denoted as RPIC_{PDAMA}) with an ability to slide even commercially available refined oils (motor oil, vegetable oil, diesel, petrol, kerosene) and complex crude oil (Supplementary Fig. 10) in addition to polar and non-polar liquids. Such solid slippery coating was subjected to post-modification with glucamine to perturb the surface chemistry. As a consequence, droplets of oils and organic solvents with low (< 30 mN/m) surface tension readily spilled on the tilted interface, as shown in Supplementary Fig. 21. The post-covalent modification of RPIC_{PDAMA} elevated the surface free energy of the coating from ~ 20 mN/m to ~ 35 mN/m (Supplementary Fig. 22). However, such chemical modification merely perturbed the surface roughness of the coating, as shown in Supplementary Fig. 22. Hence, this study suggests that the alteration of surface chemistry is sufficient to selectively compromise the sliding behaviour of low surface tension (< 30 mN/m) organic solvents and refined/crude oils, where alteration of topography is not required. Thus, the strategic modulation of chemistry provides a basis to efficiently alter the liquid sliding/spilling behaviour on a solid surface, and such principle of chemical modulation would allow to derive other functional materials in the future.”

2. The ability to “rewrite” the patterns is interesting. The authors claim that the Glu domains become buried upon heating-cooling cycle and fresh reactive MA moiety is seen at the surface. Is there a limit in terms of the number of cycles until the system becomes saturated, beyond ~ 20 cycles that the authors have done?

Response: We thank Reviewer 1 for raising this query. Yes, in this work, we have demonstrated the ability to rewrite pattern on the RPIC₂ for 20 cycles, as beyond this number of cycles the coating failed to restore the native sliding property against the low surface tension liquids. This is attributed to the limited availability of reactive MA moiety in prepared coating of thickness $\sim 20 \pm 1.6 \mu\text{m}$, where the content of PODAMA is 50 mg/cm². However, restoration ability of native anti-wetting property can be further improved just by loading more amount of reactive comb-polymer in the porous polymeric coating—during its fabrication. In this relevance, we have loaded more PODAMA in the porous coating to prepare thicker RPIC₂ to successfully demonstrate sliding/spilling cycles beyond 20 cycles.

The higher contents (100 mg/cm² and 200 mg/cm²) of PODAMA resulted in RPIC₂ with thicknesses of $\sim 51 \pm 1.8 \mu\text{m}$ and $\sim 71 \pm 1.4 \mu\text{m}$, respectively (Supplementary Fig. 24). These two different sets of RPIC₂ with higher thicknesses displayed 40 and 70 times of damage and healing cycles of the sliding property against low surface tension liquids, respectively, as shown in Supplementary Fig. 24 and 25. Actually, the higher loading of PODAMA ensures more availability of fresh MA moiety even after damage/healing cycle beyond 20 times. Although there is a limit in terms of the number of cycles of performing the healing and damage of native sliding property, it can be easily adjusted by altering the amount of infused polymer, and so the thickness of the prepared coatings. This demonstrates the versatility of the fabrication process contributes in improving the performance of the prepared slippery coating.

Supplementary Fig. 24. Plot accounting for the changes in coating thickness of reactive polymer infused coating-2 (RPIC₂) with increasing the loading amount of infused polymer polyoctadecyl acrylate maleic anhydride-2 (PODAMA₂) from 50 mg/cm² to 200 mg/cm². The error bar indicates the standard deviation with number of measurements, n=3 for each data point.

Supplementary Fig. 25. Plot demonstrating the sliding and spillage of beaded droplet of ethanol on two distinct RPIC₂ having difference in loading amount of PODMA, i.e. 100 mg/cm² and 200 mg/cm² for 40 and 70 cycles through repeating the glucamine modification and subsequently heat-treatment. The error bar indicates the standard deviation with number of measurements, n=3 for each data point.

We have incorporated the following text in the revised manuscript;

“Beyond 20 cycles, the RPIC₂ failed to restore the native sliding property against the low surface tension liquids due to the limited availability of fresh and adequate MA moiety in the prepared coating of thickness $\sim 20 \pm 1.6 \mu\text{m}$, where the content of infused PODAMA₂ is 50 mg/cm². The restoration ability of native sliding ability was further attempted to improve just by loading more reactive comb-polymer in the porous polymeric layer—during its fabrication. In this relevance, we have loaded more PODAMA₂ in the porous layer to prepare thicker RPIC₂ for successfully demonstrating the sliding/spilling cycles beyond 20 cycles.

The higher contents (100 mg/cm² and 200 mg/cm²) of PODAMA₂, resulted in RPIC₂ with a thickness of $\sim 51 \pm 1.8 \mu\text{m}$ and $\sim 71 \pm 1.4 \mu\text{m}$ and displayed reversible alteration of sliding and arbitrary spilling of low surface tension liquid on the prepared coating for 40 and 70 times, respectively as shown in Supplementary Fig. 24, 25. Actually, the higher loading of PODAMA₂ ensures more availability of fresh MA moiety even after repetitive surface modifications and followed by its erasure beyond 20 times. Although there is a limit in terms of the number of cycles for surface modification and followed by its erasure, it can be easily adjusted by varying the amount of infused polymer, thereby altering the thickness of the prepared coatings. This demonstrates the versatility of the fabrication process—which contributes to improving the performance of the prepared slippery coating.”

3. The authors claim that some of the surfaces (RPIC₄) have lower roughness and yet have larger friction force for sliding. This suggests the CAH is higher. What is the origin of the CAH? Are there variations in the surface energies that can lead to pinning points?

Response: We thank Reviewer 1 for this query. Yes, the reason behind the higher CAH for smoother interface is the elevated surface free energy. We have prepared a series of reactive co-polymer by varying the content of maleic anhydride (MA) moiety—the subsequent infusion of such polymer into the porous coating yielded coatings (RPIC_{1/2/3/4}) exhibited distinct anti-wetting behaviour as shown in Figure 2c, d, g, and Supplementary Fig. 4. To understand this phenomenon, both surface free energy and root mean square roughness (Rq) of these different coatings (RPIC_{1/2/3/4}) was estimated (Figure 2 e, f and g). The atomic force microscopic (AFM) images revealed the noticeable difference in topography of these coatings with increasing the mole fraction of maleic anhydride (MA) in the infused co-polymer, ranging from 0.13 (RPIC₁) to 0.59 (RPIC₄). A very apparent depletion in the root mean square roughness (Rq) was noticed from $\sim 48 \pm 3.7 \text{ nm}$ to $\sim 0.52 \pm 0.1 \text{ nm}$ with increasing the MA content in the selected polymers (PODAMA_{1/2/3/4}). Though it was expected that the decrease in the surface roughness would improve the sliding behaviour of the liquid droplets, thereby decreasing the CAH of liquid droplets, an opposite trend was observed. The CAH of water ($\sim 20^\circ$ to $\sim 30^\circ$) and ethanol ($\sim 9^\circ$ to 20°) droplets increases gradually with increasing the MA content in the infused co-polymers (PODAMA_{1/2/3/4}), with RPIC₄ failing to slide low surface tension liquids, instead spillage of the liquid droplet is noticed. Thus, roughness alone is not the determining factor in understanding this distinct anti-wetting phenomenon; another crucial parameter, i.e., surface free energy (SFE), also plays an important role. It was noted that SFE gradually elevates with increasing the MA content in the infused polymer (Figure 2g). RPIC₄, loaded with 0.59 mol

fraction of MA, exhibited relatively high SFE 32 ± 0.2 mN/m and displayed very high CAH, failing to slide low surface tension solvents. Therefore, SFE plays a superior role over the roughness towards the CAH of the prepared slippery coatings.

We have included following statements in the revised manuscript;

“To understand such distinct antiwetting performance exhibited by different RPIC, their surface morphologies were examined, where the mole fraction of MA moiety in selected polymers (PODAMA_{1/2/3/4}) gradually increased from 0.13 to 0.59. Atomic force microscopic (AFM) images validated the difference in the topography of prepared coatings (Supplementary Fig. 5). While RPIC₂ was embedded with dominated fibril domains (Fig. 2e), RPIC₄ displayed an ultra-smooth coating (Fig. 2f). A very apparent depletion in the root mean square roughness (Rq) was also noticed from $\sim 48\pm 3.7$ nm to $\sim 0.52\pm 0.1$ nm with increasing the MA content in the selected polymers (PODAMA_{1/2/3/4}) (Fig. 2g)”

“With increasing MA content in the infused polymer, the SFE gradually elevated (Fig. 2g). We noticed that the coating (RPIC₄) loaded with MA 0.59 mole fraction associated with relatively high SFE of $\sim 32\pm 0.2$ mN/m.”

“The CAH of water ($\sim 20^\circ$ to $\sim 30^\circ$) and ethanol ($\sim 9^\circ$ to 20°) droplets increases gradually with rising more MA moiety in the infused co-polymers (PODAMA_{1/2/3/4}) of prepared coatings (RPIC₁ to RPIC₄; Fig. 2g). The elevated SFE of the coatings because of the increasing in the content MA moiety is likely to induce more polar-polar interaction, and so higher CAH for RPIC₄. Thus, the current study experimentally validates that the SFE plays a superior role over roughness towards slippery property against low surface tension liquids and CAH of the beaded liquid droplets on the prepared coating.”

4. Related to points 1b and 3) above, when the authors claim that water droplets do not see the patterns, what happens at small tilt angle? Is there effectively a chemical domain (even a diffuse one) that leads to droplet pinning between regions with and without chemical modifications? Based on the presented work, I expect chemical domains with CA 88 and 112 degrees, which the droplet will see when the gravitational force is not large.

Response: We thank the Reviewer 1 for this concern. We are sorry for the confusion. *In our response to question # 1b, with adequate additional experimental data, we have validated that we have not exploited droplet sliding versus droplet pinning behaviour of the prepared slippery coating—rather it is a demonstration of droplet sliding versus droplet spillage phenomena. Please see the detail response against the question 1b.*

The wettability pattern in the current demonstration was prepared through spatially selective glucamine modification on the surface of RPIC₂. At a tilting angle 12° , the water droplet effortlessly slides down straight under gravity without recognizing the chemically modulated pattern on such a patterned interface (Figure 5 a, b). In contrast, a beaded droplet of ethanol recognized the pattern and selective spilled and spread on the spatially selectively glucamine-modified path to reach the other end of the patterned interface. Similarly, other polar and non-polar liquids with low surface tensions (e.g. 1-propanol, decane, 1-hexanol and dodecane) preferred to follow the pattern region (Fig. 5c, Supplementary Fig. 26), while the high surface

tension liquids (above 35 mN/m; DMF, DMSO and DIM) readily slide straight down on the surface, as shown in Fig. 5c and Supplementary Fig. 26.

Supplementary Fig. 26. Photographs showing the guided transport of low surface tension liquids (a) (decane, dodecane, 1-propanol, 1-hexanol) on the pattern interface, whereas high surface tension liquids (b) (dimethyl formamide (DMF), dimethyl sulfoxide (DMSO) and Diiodomethane (DIM)) failed to recognize the pattern—and followed a different path on the same interface to slide down.

We have repeated this study at lower tilting angle, i.e. 5° . Even at a low tilting angle, the low surface tension liquid, i.e., ethanol easily recognizes the chemically modified region and selectively spilled along the chemically modified region. But the beaded droplet of water (high surface tension liquid) behave completely different in comparison to the ethanol droplet on the same patterned interface. Despite the glucamine modification of the surface resulting in a decrease in the static contact angle value of the water droplets from $\sim 112^\circ$ to 88° , the water droplet does not spill on the tilted surface. Thus, the beaded water droplet failed to recognizing the chemically modified track, and responded differently than the beaded droplet of ethanol on the same patterned interface. Even a large droplet of water (volume of 1 ml) placed across the pattern and non-patterned region displayed effortless sliding without leaving any trace of it on the patterned interface that kept tilted at 5° as shown in Supplementary Fig. 27. Hence, the current study revalidated that the patterned interface behaved differently for low and high-surface tension liquids, even at low tilting angles.

Supplementary Fig. 27. Photographs showing the guided spillage of low surface tension liquid, i.e. ethanol along the pattern interface, whereas high surface tension liquid, i.e. water failed to recognize the pattern—and effortlessly slide down following a different path on the same interface at tilting angle of 5°.

We have incorporated the following statement in the manuscript;

“The patterned interface similarly performed even at a lower tilting angle as the chemically modified interface displayed the selective spillage of only low (< 30 mN/m) surface tension liquids, irrespective of the tilting angle (Supplementary Fig. 19). In this relevance, the patterned interface was tilted at an angle of 5° before introducing the beaded droplets of ethanol and water separately. The low surface tension liquid, i.e., ethanol, easily recognizes the chemically modified region and selectively spilled along the chemically modified region. However, the beaded water droplet (high surface tension liquid) behaved entirely differently on the same patterned interface than the ethanol droplet. Despite the glucamine modification of the surface resulting in a decrease in the static contact angle value of the water droplets from $\sim 112^\circ$ to 88° , the water droplet does not spill on the tilted surface. Thus, the beaded water droplet failed to recognize the chemically modified track, and responded differently than the beaded droplet of ethanol on the same patterned interface. Even a large droplet of water (volume of 1 ml) placed across the pattern and non-patterned region displayed effortless sliding without leaving any trace of it on the patterned interface that kept tilted at 5° as shown in Supplementary Fig. 27. Hence, the current study revalidated that the patterned interface behaved differently for low and high-surface tension liquids, even at low tilting angles.”

Responses to Reviewer #2

Dhar and colleagues present a clever approach via selective modification of a (semi)crystalline polymer network such that during a thermally-mediated phase transition, the network reverts to its native wettability. The authors present a compelling narrative for reversible wettability, but the process is not quite fully reversible. After

treating the surface with glucamine, a ring opening reaction is triggered with the maleic anhydride monomers that have co-polymerized onto the polymer chains next to neighboring, varying length n-alkyl side chains off of the acrylates in the chain backbone. While heating and cooling the material does "erase" the crystallinity, it does not seem that the glucamine is removed from the polymer network, rather the side chain semi-crystallinity is disrupted. Over repeat cycles, the accessible maleic anhydride will become less and less leading to a system that is no longer able to be further surface modified due to using up the available MA in the network.

Response: We thank the reviewer for the high recognition and constructive suggestions of our work. In light of the referee's comments, we have worked hard to revise the manuscript.

The authors claim rewrite-ability over "multiple" cycles, which is likely largely dependent on the relative ratio of Glu to MA (or the ink density per rewrite cycle). The authors would do well to bound these statements if not experimentally at least with a theoretical discussion of the limits of rewrite-ability, which are likely related to this ink density metric per writing cycle. The authors could also perhaps imagine or suggest future work where a thermally labile reaction could replace the covalent Glu-MA reaction and perhaps form a more reuseable surface.

Response: We thank reviewer 2 for proving important and constructive suggestion. Yes, in this current work, the prepared coating (RPIC₂) to rewrite the pattern for only 20 cycles, as beyond this number of cycles the coating failed to restore the native sliding property against the low surface tension liquids. This is attributed to the limited availability of reactive MA moiety in the prepared coating of thickness $20 \pm 1.6 \mu\text{m}$, where the content of PODMA is 50 mg/cm^2 . However, the re-writing ability of the pattern can be further improved just by loading more amount of reactive comb-polymer in the porous polymeric coating—during the process of its fabrication to enable adequate chemical reactions between glucamine (Glu) and maleic anhydride (MA) for more cycles. During the heating-cooling cycle, the reacted Glu domains become buried, and fresh reactive MA moiety appears on the surface. This process enables the erasing the chemical modification and allows for rewriting another pattern for more cycles. In this relevance, we have loaded more PODAMA in the porous coating to prepare thicker RPIC₂ to successfully demonstrate the erasing/rewriting cycles beyond 20 times.

It is worth to mention that copolymer with increased mole fraction of MA provided coating (RPIC₄) with inability to slide low surface tension liquids—even before association of glucamine modification as shown in Figure 2d,g.

The higher contents (100 mg/cm^2 and 200 mg/cm^2) of PODAMA resulted in RPIC₂ with thicknesses of $\sim 51 \pm 1.8 \mu\text{m}$ and $\sim 71 \pm 1.4 \mu\text{m}$, respectively. These two different sets of RPIC₂ with higher thicknesses displayed 40 and 70 times of damage and healing cycles of the sliding property against low surface tension liquids, respectively, as shown in Supplementary Fig. 24, 25. Actually, the higher loading of PODAMA ensures more availability of fresh MA moiety even after damage/healing cycle beyond 20 times. Although there is a limit in terms of the number of cycles of performing the healing and damage of native sliding property, it can be easily adjusted by altering the amount of infused polymer, and so the thickness of the prepared coatings. This demonstrates the versatility of the fabrication process contributes in improving the performance of the prepared slippery coating.

Supplementary Fig. 24. Plot accounting for the changes in coating thickness of reactive polymer infused coating-2 (RPIC₂) with increasing the loading amount of infused polymer polyoctadecyl acrylate maleic anhydride-2 (PODAMA₂) from 50 mg/cm² to 200 mg/cm². The error bar indicates the standard deviation with number of measurements, n=3 for each data point.

Supplementary Fig. 25. Plot demonstrating the sliding and spillage of beaded droplet of ethanol on two distinct RPIC₂ having difference in loading amount of PODMA, i.e. 100 mg/cm² and 200 mg/cm² for 40 and 70 cycles through repeating the glucamine modification and subsequently heat-treatment. The error bar indicates the standard deviation with number of measurements, n=3 for each data point.

In this current study, the rewritable pattern on slippery coating is derived through a stable covalent bonding between glucamine and available MA moiety of the co-polymer—and thus the limited availability of MA moiety allowed rewriting the pattern for a certain number of cycles. To address this issue, spatially selective chemical modification may be attempted following a dynamic covalent bonding as an alternative approach. So that the chemical modification can be erased in the presence of specific and appropriate stimuli, such as pH, temperature, UV light, etc., and the same interface can be reutilized for rewriting the pattern. For example, an essential chemical modification can be achieved through associating imine bond—which is known to be labile towards acidic hydrolysis. Such an approach would allow the pattern to be rewritten for several cycles as the same reactive moiety is reversibly consumed and subsequently reused to alter the surface free energy. Such strategy will be explored separately in future studies.

We have incorporated the following text in the revised manuscript;

“Beyond 20 cycles, the RPIC₂ failed to restore the native sliding property against the low surface tension liquids due to the limited availability of fresh and adequate MA moiety in the prepared coating of thickness $\sim 20 \pm 1.6 \mu\text{m}$, where the content of infused PODAMA₂ is 50 mg/cm². The restoration ability of native sliding ability was further attempted to improve just by loading more reactive comb-polymer in the porous polymeric layer—during its fabrication. In this relevance, we have loaded more PODAMA₂ in the porous layer to prepare thicker RPIC₂ for successfully demonstrating the sliding/spilling cycles beyond 20 cycles.”

“The higher contents (100 mg/cm² and 200 mg/cm²) of PODAMA₂, resulted in RPIC₂ with a thickness of $\sim 51 \pm 1.8 \mu\text{m}$ and $\sim 71 \pm 1.4 \mu\text{m}$ and displayed reversible alteration of sliding and arbitrary spilling of low surface tension liquid on the prepared coating for 40 and 70 times, respectively as shown in Supplementary Fig. 24, 25. Actually, the higher loading of PODAMA₂ ensures more availability of fresh MA moiety even after repetitive surface modification and followed by its erasure beyond 20 times. Although there is a limit in terms of the number of cycles for surface modification and followed by its erasure, it can be easily adjusted by varying the amount of infused polymer, thereby altering the thickness of the prepared coatings. This demonstrates the versatility of the fabrication process—which contributes to improving the performance of the prepared slippery coating.”

“In this current study, the rewritable pattern on slippery coating is derived through a stable covalent bonding between glucamine and available MA moiety of the co-polymer—and thus the limited availability of MA moiety in the prepared coating allowed rewriting the pattern for a certain number of cycles. To address this issue, spatially selective chemical modification may be attempted following a dynamic covalent bonding as an alternative approach. So that the chemical modification can be erased in the presence of specific and appropriate stimuli, such as pH, temperature, UV light, etc. to recover the native functional group, and the same interface can be reutilized for rewriting the pattern. For example, an essential chemical modification can be achieved through associating imine bond—which is known to be labile towards acidic hydrolysis. Such an approach would allow the pattern to be rewritten for several cycles as the same reactive moiety is reversibly consumed and subsequently reused to alter the surface free energy. Such strategy will be explored separately in future studies.”

Nonetheless, these critiques should not diminish from an excellent story, compelling videos and well-engineered figures both within the main text and in the supplementary information provided.

The work is detailed and likely reproducible in a well-maintained chemistry or polymer science laboratory. I applaud the authors for interesting and well-documented work that could lead to other interesting avenues in the field in the years to come.

Response: We thank Reviewer 2 for appreciating and recognizing the potential of the current work.

REVIEWERS' COMMENTS

Reviewer #1 (Remarks to the Author):

I have considered the revised manuscript by Dhar et al. entitled 'Rewritable' and 'Liquid-specific' recognizable wettability pattern.

I highly appreciate the additional works and demonstrations that the authors have carried out to address my concerns. In particular, their demonstrations of using additional liquids and another polymer substrate resolve my main concerns. I find these are quite convincing now. It is also nice that the authors carried out additional experiments with different tilting angle and droplet volume, and the experiments demonstrating larger number of cycles is interesting.

With the additional results and changes to the manuscript, I am now happy to suggest this manuscript to be accepted for publication in Nature Communications.

Reviewer #2 (Remarks to the Author):

The authors have been extremely responsive to the reviewers and in the current version do not hype their results, but provide solid evidence to bound their work in several key areas such as concerning re-use and sliding angles. They address the realistic bounds of the technology and chemistry which could be very insightful for future scholars within the field.

I recommend accepting the work given the modifications and clarifications made in the re-submission.

Point-by-point response to the reviewer's comment

Responses to Reviewer 1:

Reviewer #1:

I have considered the revised manuscript by Dhar et al. entitled 'Rewritable' and 'Liquid-specific' recognizable wettability pattern.

I highly appreciate the additional works and demonstrations that the authors have carried out to address my concerns. In particular, their demonstrations of using additional liquids and another polymer substrate resolve my main concerns. I find these are quite convincing now. It is also nice that the authors carried out additional experiments with different tilting angle and droplet volume, and the experiments demonstrating larger number of cycles is interesting.

With the additional results and changes to the manuscript, I am now happy to suggest this manuscript to be accepted for publication in Nature Communications.

Response: We are thankful to the reviewer for appreciating the work and recommending the revised manuscript for publication in Nature Communications.

Responses to Reviewer #2

The authors have been extremely responsive to the reviewers and in the current version do not hype their results, but provide solid evidence to bound their work in several key areas such as concerning re-use and sliding angles. They address the realistic bounds of the technology and chemistry which could be very insightful for future scholars within the field.

I recommend accepting the work given the modifications and clarifications made in the re-submission.

Response: We are grateful to the reviewer for recognizing the potential of the work and recommending the revised manuscript for publication in Nature Communications.